

# High-precision ID-TIMS Cassiterite U-Pb systematics using a low-contamination hydrothermal decomposition: implications for LA-ICP-MS and ore deposit geochronology

Simon Tapster[1*], Joshua W. G. Bright[1,2]

1. Geochronology and Tracers Facility, British Geological Survey, Keyworth, Nottinghamshire, NG12 5GG, UK

2. Department of Chemistry, University of Surrey, Guildford, GU2 7XH, UK

*Email: simont@bgs.ac.uk



**1. Abstract**

Cassiterite ($SnO_2$) is the most common ore phase of Sn. Typically containing 1-100 µg/g U and relatively low concentrations of common Pb the mineral has been increasingly targeted for U-Pb geochronology, principally via micro-beam methods, to understand the timing and durations of granite related magmatic-hydrothermal systems throughout geological time. However, due to the extreme

resistance of cassiterite to most forms of acid digestion, to date, there has been no published method permitting the complete, closed system decomposition of cassiterite under conditions where the basic necessities of measurement by isotope dilution can be met, leading to a paucity of reference, and validation materials. To address this a new low blank (<1 pg Pb) method for the complete acid decomposition of cassiterite utilising HBr in the presence of a mixed U-Pb tracer, U and Pb purification,

and TIMS analyses has been developed. Decomposition rates have been experimentally evaluated under a range of conditions. A careful balance of time and temperature is required due to competing effects (e.g. HBr oxidation) yet decomposition of 500 µm diameter fragments of cassiterite is readily achievable over periods comparable to zircon decomposition. Its acid resistant nature can be turned into an advantage, by leaching common Pb-bearing phases (e.g. sulfides, silicates) without disturbing the U-

Pb systematics of the cassiterite lattice. The archetypal Sn-W greisen deposit of Cligga Head, SW England, is used to define accuracy relative to CA-ID-TIMS zircon U-Pb ages and demonstrate the potential of this new method, for resolving high resolution timescales (<0.1 %) of magmatic-hydrothermal systems. However, analyses also indicate that the isotopic composition of initial common Pb varies significantly, both between crystals and within a single crystal. This is attributed to significant

fluid-rock interactions and the highly F-rich acidic nature of the hydrothermal system. At micro-beam precision levels, this issue is largely unresolvable and can result in significant inaccuracy in interpreted ages. However, this method can, for the first time, be used to properly characterise suitable reference materials for micro-beam cassiterite U-Pb analyses, thus improving the accuracy of the U-Pb cassiterite chronometer as a whole.



## 2. Introduction

Cassiterite ($SnO_2$) is the primary tin ore, precipitated from magmatic-hydrothermal fluids exsolved in association with reduced granitic systems and pegmatite bodies. The mineral systems that contain cassiterite are globally distributed and they span the majority of geological time from the Archean to the Neogene (Kesler and Wilkinson, 2015). Cassiterite commonly contains 1 to 100 µg/g U and has the potential to contain relatively low levels of common Pb (e.g. Li et al., 2016; Moscati and Neymark, 2019; Neymark et al., 2018). The potential of cassiterite as a U-Pb geochronometer to understand the timing of magmatic-hydrothermal Sn deposits, and their many associated elements critical for modern technology e.g. Li, W, Nb, Ta, has long been recognised since the first reported Isotope Dilution Thermal Ionisation Mass Spectrometry (ID-TIMS) U-Pb analyses using  (Gulson and Jones, 1992). Yet research into cassiterite U-Pb geochronology paused for some time due to the questionable accuracy of the first published results (McNaughton et al., 1993) and the difficulty in the hydrothermal decomposition of cassiterite required for isotope dilution methods due to the extremely acid resistant nature of cassiterite.

With the advent of micro-beam techniques, cassiterite U-Pb geochronology has now become widely accessible. Techniques such as LA-ICP-MS and Ion microprobe (Carr et al., 2017) offer a rapid solution to analysing the U-Pb systematics of cassiterite without the direct need for dissolving a sample.

With analytical precision of ~1-2 % 2σ, and a current total uncertainty limit of ~2 % based upon long-term reproducibility and inter-laboratory comparison of zircon U-Pb analyses (Horstwood et al., 2016), the LA-ICP-MS U-Pb techniques offer the capability to provide chronology that may address issues such as the timing of ore deposition at broad, regional scales. Issues specific to cassiterite LA-ICP-MS U-Pb analyses are outlined below.  Higher temporal resolution questions, such as those that begin to address processes on magmatic timescales require a refinement of these uncertainties. For micro-beam techniques this may result from the refined standardisation of the methodology and identification of systematic uncertainties identified over long term analysis using well characterised reference materials (RMs)(Horstwood et al., 2016).



Accuracy of U-Pb micro-beam geochronology requires that effects such as matrix interaction with analyte ion beams and "down-hole" inter-element fractionation can be corrected for. This is commonly

negated by using RMs with an equivalent chemistry and ablation (or sputtering) characteristics (i.e. a matrix matched RM). Samples of unknown isotopic composition are measured relative to well characterised RMs that are relatively homogenous at the scale of the spatial resolution and at the precision of the analyses and typically characterised by ID-TIMS U-Pb analyses. .

Thus far, a fundamental issue for cassiterite U-Pb geochronology arises from the challenges in

achieving total decomposition of cassiterite, in a single closed system step, that is required for the effective quantitative U/Pb determination of the material by isotope dilution methods. Previous ID-TIMS U-Pb studies (Gulson and Jones, 1992; Rizvanova et al., 2017; Yuan et al., 2008) utilising concentrated HCl required stepwise decomposition and acid recharges, that still only achieved partial decompositions in many instances.  These studies also required the post-decomposition addition of

isotopic U-Pb tracers. This approach will likely preserve the Pb isotope systematics of the sample, for cassiterite this will represent a mixture of common Pb (Pbc) components and radiogenic Pb (Pb*). However, it introduces a high potential to fractionate the U-Pb inter-element ratios, either due to the incongruous dissolution of U and Pb, and different elemental behaviours before equilibrium with the tracer is achieved. These issues will in turn affect the integrity of U-Pb derived dates and therefore the

interpretation of Pb loss, or Pbc-Pb* mixing trajectories that require assumptions about concordance of the U-Pb chronometers. As noted by Neymark et al. (2018) these effects lead to reverse discordance (U loss or Pb gain) between the resulting cassiterite analyses. Whilst this effect can be easily detected in older cassiterite data where trajectories approach being perpendicular to Concordia, within younger samples, fractionation will merely drive samples near parallel to the line of Concordia, making it

difficult to distinguish from real age variations.

Many previous micro-beam cassiterite U-Pb studies have used some combination of zircon RM or glass RM, with a "matrix matched" cassiterite RM that has had isotopic ratios characterised by ID-TIMS following a stepwise decomposition (e.g. Yuan et al., 2011, 2008; Zhang et al., 2017; Li et al., 2016;





Liu et al., 2007). The reference values have the potential to have been effected by the issues described

above, with inaccuracies propagating into the resulting ages.

To overcome the matrix matching issues that stem from the absence of well characterised cassiterite material that can be used as a RM, Neymark et al. (2018) developed an approach that utilised the highly repeatable ablation characteristics of cassiterites, to normalise the matrix biases relative to a NIST glass primary RM by deriving a "fractionation factor" for each analytical session. This was achieved by

measuring the offset between the lower intercept age on a Terra Wasserburg (T-W) plot, and an assumed "true" age defined by the Pb-Pb isochron that was shown to be in good agreement with independent chronological constraints for the Proterozoic "SPG" cassiterite. The application of this approach to a wide range of materials across geological time, and the general agreement with independent geological and temporal constraints of their associated systems (Moscati and Neymark, 2019; Neymark et al.,

2018) indicates that this approach was both versatile, and can be reliably applied to constrain the general timing of ore deposits. However, without characterising, and understanding, the U-Pb systematics of cassiterite materials by precise methods the level of accuracy of the methods, and the potential interpretive resolution for geological problems are difficult to assess.

Regardless of the micro-beam approach adopted, it is crucial that the session and long term accuracy

and dispersion of any U-Pb analytical approach can also be validated by the analysis of independently well characterised cassiterites (i.e. secondary and tertiary RMs) (e.g. Horstwood et al. 2016).

The motivation for developing a method for cassiterite ID-TIMS U-Pb geochronology that involves the total decomposition of cassiterite in the presence of, and achieving equilibrium with a U-Pb isotope tracer, is to provide a means to characterise cassiterite materials to assess accuracy, and a means to

examine magmatic-hydrothermal timescales within cassiterite bearing deposits beyond the consensus ca. 2 % absolute uncertainty that is permitted by micro-beam U-Pb methods (e.g., Horstwood 2016).

Desirable characteristics of a method are also: 1) contribution of  low amounts of Pbc to the sample during decomposition and chemical separation using  reagents that can be easily handled and distilled; 2) that the method can achieve decomposition on timescales that are easily operable for the laboratory

environment e.g. zircon decomposition is typically achieved within ~72 hours; 3) That the method





utilises geologically meaningful amounts of material, capable of resolving spatial variations in isotopic ratios.

This study demonstrates the potential of cassiterite decomposition with concentrated HBr to fulfil the necessary criteria for routine ID-TIMS cassiterite U-Pb geochronology. We provide preliminary data

for two cassiterite materials that have the potential for use as RMs, and we evaluate the accuracy of cassiterite U-Pb geochronology using the case study of the classic W-Sn magmatic hydrothermal system of Cligga Head, SW England, which illustrates the complexities and biases that may arise for cassiterite U-Pb geochronology.

**3. Previous methods for the decomposition of Cassiterite**

Cassiterite is notoriously resistant to decomposition in acids (Mathur et al., 2017; Yamazaki et al., 2013). Although analysis of Sn isotopes within cassiterite is well established, decomposition has typically been conducted by alkali fusion (Hall, 1980; Sear, 1997), reduction with graphite (Clayton et al., 2002; Hall, 1980) or potassium cyanide reduction (Haustein et al., 2010; Mathur et al., 2017). These

methods require high temperatures (800-1200°C) and open systems, making them inappropriate for ID U-Pb geochronology.

Hydroiodic acid has been demonstrated as an effective means of cassiterite decomposition at relatively low temperatures 100°C (Caley, 1932). More recent work was able to dissolve 1 mg of cassiterite in high pressure decomposition vessels at 100°C over four sequences of 24 hours, exchanging the acid at

each step (Yamazki et al., 2013; Mathur et al., 2017). Despite the potential of HI as a means to decompose cassiterite, there are several drawbacks as HI readily reacts with oxygen within air requiring refrigerated storage, and at the time of writing was less readily available with certification to <1 pg/ml Pb and was significantly more expensive when compared with HBr. Likewise, the decomposition procedure of Yamazki et al. (2013) yielded a 1 ng Sn blank, and whilst it is conjecture to predict the

associated Pbc blank, if on this order of magnitude it would rule out HI as a low contamination method for high-precision geochronology of cassiterite. Furthermore, upon drying of the dissolved samples Yamazki et al. (2013) noted the formation of precipitates which were insoluble with a range of acids



(HCl, HF, HClO$_3$, HNO$_3$), it is possible that these could also sequester Pb and U, significantly decreasing recoveries.

Previous efforts to analyse cassiterite U-Pb through ID methods for geochronology have utilised a concentrated HCl based decomposition within high-pressure acid digestion vessels (Gulson and Jones 1992; Liu 2007; Yuan et al., 2008; Yuan et al., 2011; Rizvanova et al., 2017). The previously reported contributions of Pbc from the method have been have been on the order of 0.2-0.3 ng (Gulson and Jones, 1992) to 0.8 ng (Yuan et al., 2011, 2008), many orders of magnitude greater than what is typically

achieved by modern zircon CA-ID-TIMS geochronology methods. The issues of incomplete decomposition in previous studies was highlighted by Neymark et al. (2018). Although not explicit about the number of fractions that did not dissolve, the methods of Yuan et al. (2008) indicate that at least some of their cassiterite was not decomposed after 72 h in 12 M HCl at 205°C in high-pressure digestion vessels, and required multiple acid exchange steps, and also further crushing with a pestle and

mortar in-between decomposition steps. Gulson and Jones (1992) also utilised a 48-72 h, ~200°C, 12 M HCl decomposition stage within a high pressure digestion vessel and noted that in some instances for ~200 Ma cassiterite, <5 % of cassiterite was dissolved after numerous steps.

The potential for inaccuracy in cassiterite U-Pb geochronology was illustrated by the study of Gulson and Jones (1992) who presented a very precise age (± ~0.1 % 2σ) from a discordia regression line with

an upper intercept age of 2098.6 Ma ± 3.1 Ma for cassiterite associated with granitic rocks within the Bushveld Complex, South Africa. This cassiterite age was quickly challenged at the time on account of much younger zircon ages of the associated granites (McNaughton et al., 1993). Extensive high-precision CA-ID-TIMS zircon U-Pb geochronology of the relatively older Rustenberg layered igneous suite of the Bushveld Complex (Mungall et al., 2016; Scoates and Friedman, 2008; Zeh et al., 2015)

has now confidently demonstrated that the cassiterite age is at least > 40 Myrs (>2 %) too old.

The potential to decompose tin oxides at pressure using 9 M (48 %) HBr in high pressure digestion vessels has been identified for some time now (Doležal et al., 1969), yet its application has yet been somewhat overlooked for U-Pb geochronology despite its strong acidic, reducing nature that is comparable to HI.






**4. Controls on the timescales of cassiterite hydrothermal decomposition with 9M HBr**

The ability of 9 M HBr to decompose Cassiterite over a range of temperatures and grain sizes was investigated in a series of step-leaching experiments. Timescales were assessed for two different cassiterite crystal samples (Fig. 1): 1) "Casnig" with an age of ~150 Ma and a heavily quenched

Cathodoluminesence (CL) response, indicating a high concentration of Fe (Farmer et al., 1991); 2) A single ~1 cm diameter cassiterite from the Permian Cligga Head, SW England which demonstrates strong zonation under CL. Around 0.015 g to 0.03 g of cassiterite fragments, both powdered in a pestle and mortar to <10 µm to 50 µm, and as coarse 500 µm length cubes, were decomposed in a 50x molar excess of 9 M HBr in Savillex high pressure digestion vessels (Parr Bombs) at a range of temperatures

180°C-230°C. Decomposition steps lasted for 12 h, or 96 h, and the mass decrease in the residual cassiterite was documented (see supplementary material for full data), following extraction and extensive rinsing of the leachate. The acid was recharged and the experiment repeated. Complexity in the weighing arose from $Br_2$, released during the decomposition steps, diffusing into the PFA vessels thus increasing the mass. The mass would then decrease as $Br_2$ diffused out of the PFA vessel during

hot plate dry down steps (120°C) before weighing. Experimental runs of PFA vessels without cassiterite present indicated the potential mass variation at each weighing stage to be ±0.00154 g (2s). Scanning Electron Microscope imagery and Energy Dispersive Spectroscopy of the precipitated leachate after a decomposition step identified crystalline tin bromide salts, and spongy textured Ti, Nb and Fe bromides which reflect minor elements contained within the cassiterite (Fig. 1). No tin oxide was identified

supporting that the mass reduction at each stage was due to decomposition of cassiterite in HBr.



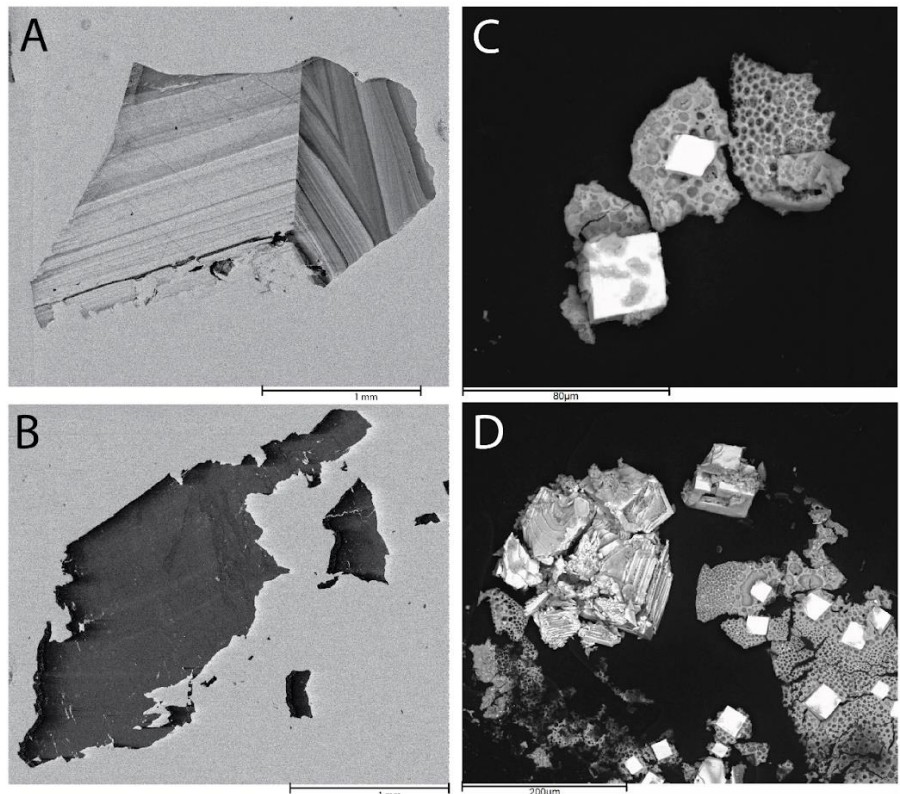

**Figure 1.** A) Representative CL image of cassiterite from Cligga Head used for decomposition experiments and geochronology; B) Representative CL image of the Casnig cassiterite used for decomposition experiments; C and D) SEM images of the precipitate formed after drying down the HBr

leachate after cassiterite decomposition. Light coloured crystals are tin bromide crystals, the "spongey" texture contains bromides of Fe, Nb and Ti.

Decomposition experiments run at 180°C and 200°C did not exhibit a significant reduction in mass

beyond the mass variation in the beaker due to $Br_2$ diffusion. No clear temperature control was exerted between 210°C and 230°C and most experiments demonstrated a relatively large variation in the amount dissolved over 12 h from step to step. Likewise, there was no consistent variation in the amount decomposed over 12 h between the two different starting materials of cassiterite at a given temperature.





In general more cassiterite decomposed when powdered in a pestle and mortar than when placed in as

larger fragment, although significant variation in the rate of a given step was still present.

Notably there was little variation between the total mass of cassiterite decomposed in either a 96 h step

or a 12 h step, indicating the majority of cassiterite decomposition occurs within the first 12 h or less.

We suggest this apparent decrease in rate resulted from a rapid reduction in HBr molarity when HBr

was exposed to high temperatures. The strong brown discolouration and mass increase of the

decomposition vessel that was not unique to cassiterite bearing experiments, implied that $Br_2$ was

generated likely through a reaction with atmospheric oxygen. It is well documented that aqueous

solutions of HI readily react with atmospheric oxygen at room temperatures to produce $I_2$, which further

react with HI to produce $HI_3$, and a similar reaction between $O_2$ and HCl occurs, just at a much slower

rate (and requiring higher temperatures). It follows that this reaction occurs with HBr, although it

appears to not be previously well documented.

The decomposition experiments act as an informative guide to cassiterite decomposition, but given the

scales of the experiment design required to overcome weighing issues, it is unlikely they

comprehensively reflect the rates of cassiterite decomposition in 9M HBr. Whether a cassiterite sample

completely decomposes within a single step is likely to be fundamentally controlled by the surface area

to volume (or mass) ratio of the cassiterite that is exposed to the acid during the earliest stages of

decomposition, before the HBr is lost and becomes ineffective. Use of a significant excess of

concentrated HBr is proposed, in the later experiments for U-Pb geochronology a molar excess of ~190

times was used. To ensure complete single step decomposition can be achieved, we advise carrying out

a trial decomposition experiment to ensure a given mass and fragment size of a specific cassiterite

sample can be decomposed within the parameters of the experiment chosen.





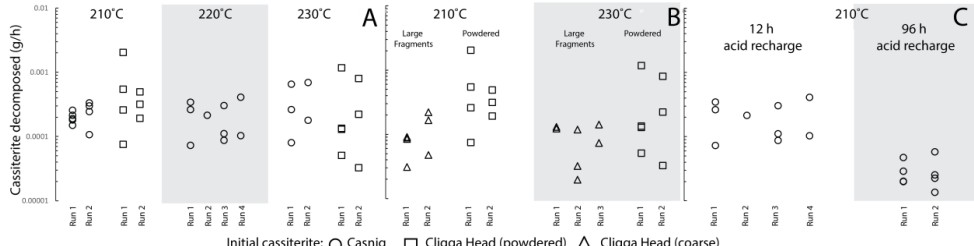

**Figure 2.** Plots of integrated decomposition rates for different initial start cassiterite materials. Variations in mass below those induced by $Br_2$ diffusion into the beaker are not displayed  A) Shows variation of rate over a 12 h decomposition step at different temperatures within high pressure vessels for powdered cassiterite. B) Shows differences in rates between coarse fragments approximating cubes 500 µm in length, and cassiterite powdered in a pestle and mortar at 210°C and 230°C. C) Illustrates the difference in rate averaged over 12 h to 96 h decomposition steps at 210°C. See text for discussion.

## 5. Method for Cassiterite ID-TIMS U-Pb geochronology

### 5.1 Sample preparation

Fragments of cassiterite grains were crushed under acetone in a pre-cleaned agate pestle and mortar, and transferred to a PFA beaker.  The acetone was then evaporated and the powdered material was rinsed in 4 M $HNO_3$ and refluxed in aqua regia at 120°C with the aim of dissolving sulfide and Fe-oxides inclusions that were exposed to the surface of the powdered grains (Gulson and Jones, 1992). The sample was rinsed in $H_2O$, and in the case of samples that underwent HF leaching, refluxed at 120°C overnight in 29 M HF. Sub-aliquots visually approximating the material contained in ~500 µm sided cubes of the leached powdered cassiterite were transferred into 500 µl PFA microcapsules (Parrish, 1987) that had undergone several steps of overnight cleaning in 9 M HBr within the Parr vessels, in addition to normal HF and HCl cleaning routines. Samples were refluxed and rinsed in 1 M HBr before being spiked with ~0.005 g of EARTHTIME 535 tracer (Condon et al., 2015).  Approximately 450 µl of 9 M HBr (Romil Ultra Purity Acid, UPA analysed to have <0.04 pg/ml Pbc) was added before placing in the oven for 5 nights at 210°C. Following decomposition, the solution was dried on a hot plate at 120°C and converted to a chloride by the addition of 400 µl of 3 M HCl and a further 12 hours at 180°C.



One noteworthy observation made over the course of cleaning the PFA capsules in 9M HBr overnight at 230°C was that the apparent Pbc was found to have doubled relative to the previous cleaning stage at 210°C, and the equivalent amount of in-house [208]Pb tracer produced very poor beam intensities. This

was interpreted as a result of organic compounds released by the reaction of the HBr with the PFA capsule. Despite potential benefits from more rapid decomposition at 230°C, lower temperatures are recommended.

**5.2 Cassiterite U-Pb anion exchange chromatography**

Following investigation isolating Pb and U from matrix elements using  a homogeneous solution of dissolved cassiterite equating to a ~500 µm length cube grain (or 0.875 mg at an assumed density of ~ 7 g/cm$^3$) and  0.05 ml of pre-cleaned AG 1-x8 200-400 mesh resin (BIO-RAD, CA, USA), a two stage column chemistry procedure was determined. In the first stage a modification of the HCl based anion exchange chromatography typically used for zircons (Krogh, 1973) was used. The sample was loaded

onto the columns in 3 M HCl, before further 3M HCl  being added in several stages of 30 µl and then several stages of 130 µl 3 M HCl. These washes contained the majority of eluted Nb, Ti and Th. The Pb was eluted in three stages of 150 µl 6M HCl, collected, and dried down with ~10 µl 0.03 M $H_3PO_4$.. The U was eluted using three steps of 150 µl of $H_2O$, followed by one step of 150 µl of 1M HCl. As the U fraction was found to contain almost all eluted Fe and Sn, a further U clean up stage was required.

The U-bearing solution was dried and re-dissolved in 300 µl 8 M $HNO_3$ before being loaded on to columns, again containing 0.05 ml of pre-cleaned AG 1-x8 resin.  Columns were washed in three stages of 350 µl 8 M $HNO_3$ followed by two steps of 350 µl 8 M HCl. The U was eluted using 2 steps of 200 µl 0.2 M HCl, collected, and dried down with ~10 µl 0.03 M $H_3PO_4$.

The Pbc procedural blank, for steps after crushing in a pestle and mortar, was found to be on the order

of ~0.5 pg. The procedural blank will be subject to laboratory variations, and the efficacy with which any material abraded during crushing within a pestle and mortar can be removed prior to decomposition within HBr.  The effects of HF leaching prior to decomposition are evaluated below, the lowest total sample and laboratory Pbc of ~1.5 pg including any pestle and mortar contributions indicates that



crushing is a minor contributor post-HF leaching. Crucially the component of Pbc contributed by the

method is one to four orders of magnitude less than the amount of Pb* of the materials examined within

this study, and the initial Pbc within the cassiterite is likely one or two orders of magnitude greater than

that from the laboratory.

### 5.3 Mass spectrometry and data reduction

The Pb and U of a given sample were independently loaded on a zone-refined Re filament in 1.5 µl of

silica gel matrix (Gerstenberger and Haase, 1997). Isotope ratio measurements were made using a

Thermo Triton TIMS at the British Geological Survey) following typical methods described by Tapster

et al. (2016). Raw U and Pb data were filtered using the Tripoli software program (Bowring et al.,

2011). Data reduction and uncertainty propagation used a modified excel spreadsheet (Schmitz and

Schoene, 2007). Concordia diagrams and regressions were constructed using the Excel add-in

'ISOPLOT 4.15' (Ludwig, 2008) and initial disequilibrium corrections utilised the IsoplotR package

(Vermeesch, 2018)

## 6. Cassiterite ID-TIMS U-Pb results


### 6.1 SPG-IV ad hoc RM Cassiterite

The SPG-IV cassiterite is taken from the Pitkäranta ore district, Russian Karelia, and was utilised by

Neymark et al. (2018) as a cassiterite reference material for deriving the "fractionation factor" of

analytical sessions. The age of the cassiterite based on geological constraints and zircon dating of

associated magmatism was inferred between 1546.7 to 1537.9 Ma (Amelin et al., 1997). Neymark et

al. (2018) presented a LA-ICP-MS $^{207}$Pb*/$^{206}$Pb* weighted mean  date of 1542.7 ± 1.5 Ma and a date of

1539.8 ± 3.5 Ma for three partially dissolved, reversely discordant ID-TIMS analyses (N. Rizvanova,

written communication, 2017 in Neymark et al. 2018).

The free regression of three tightly clustered ID-TIMS analyses (Fig. 3) of spatially independent

aliquots of the SPG-IV cassiterite yields a lower intercept of 1535.9 ± 5.5 Ma (MSWD = 2.0).  As




utilised by Neymark et al. (2018) for dates that cluster close to Concordia, anchoring the common Pb component of the T-W ischron to the mean $^{207}Pb/^{206}Pb$ value of 11 galena analyses from the ore system $(1.0104 \pm 0.0062)$ (Larin et al., 1990) generates a lower intercept date of $1536.6 \pm 1.0$ (MSWD = 1.02). The total Pbc amounts for these analyses ranges from ~5 pg to 15 pg and $^{206}Pb/^{204}Pb$ range from ~12000

to ~23000.

Comparison of the ID-TIMS T-W lower intercept dates with the LA-ICP-MS Pb-Pb isochron dates used to derive the fractionation factor (Neymark et al., 2018) indicates an absolute offset ~0.4 % older for the LA-ICP-MS Pb-Pb isochron data. The equivalent ID-TIMS $^{204}Pb/^{206}Pb$-$^{207}Pb/^{206}Pb$ isochron date for this study yields $1540.9 \pm 3.6$ Ma that is broadly in concert with that derived by the LA-ICP-MS data

(Neymark et al., 2018).

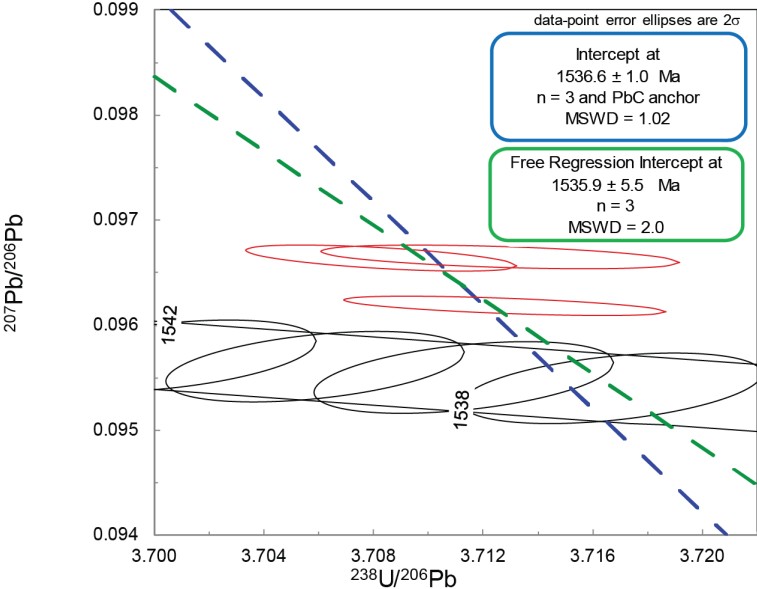

**Figure 3.** T-W plot of SPG-IV cassiterite showing lower intercept dates for freely regressed data (green

line), and regression anchored to the Pbc $^{207}Pb/^{206}Pb$ of Galena analyses from the ore system (blue line).





### 6.2 Jian-1 Cassiterite

The Jian-1 cassiterite is derived from the Jiangxi W–Sn district, South China (Neymark et al., 2018).

Previously a LA-ICP-MS cassiterite weighted mean $^{206}$Pb/$^{238}$U date of 159.5 ± 1.5 Ma (n = 31; MSWD = 0.4) (Zhang et al., 2017) was reported for the deposit. Superseded by a weighted mean $^{206}$Pb/$^{238}$U date of 156.55 ± 0.36 Ma (n = 40; MSWD=1.4) (Neymark et al. 2018). In both studies data points overlapping Concordia within uncertainty and are interpreted as concordant, or free of Pbc.

Four individual aliquots of the Jian-1 Cassiterite were analysed by ID-TIMS. All data are discordant on

the T-W plot (Fig. 4), and are interpreted as containing a component of initial common Pb. The total Pbc amounts for these analyses ranges from ~1.5 pg to 14 pg and their $^{206}$Pb/$^{204}$Pb range from ~140 to ~5300. The three data points with greatest $^{206}$Pb/$^{204}$Pb form a lower intercept of 155.05 ± 0.20 Ma (MSWD = 2.1), in agreement with the lower intercept formed by all four data points of 154.969 ± 0.082 Ma (MSWD = 1.4).

Comparison with LA-ICP-MS data (Fig. 4) indicate the coarser-scale sampling of ID-TIMS analyses identified domains with lower Pb*/Pbc relative to the previously published LA-ICP-MS data (Neymark et al., 2018) and therefore provide additional spread and an isochron on the T-W plot, that seemingly was not permitted by the LA-ICP-MS data. Despite this the isochron overlaps within uncertainty with the majority of previously published LA-ICP-MS data for Jian-1 (Neymark et al., 2018), indicating that

the accuracy of the measurements by micro-beam was relatively robust at the 1-3 % precision of single data points. However, the ID-TIMS lower intercept date of 154.969 ± 0.082 Ma is ~1 % younger than the LA-ICP-MS weighted mean $^{206}$Pb/$^{238}$U date of 156.55 ± 0.36 Ma (n = 40; MSWD = 1.4) (Neymark et al., 2018). This offset would likely reduce further if a fractionation factor was renormalised to the ~0.4 % younger ID-TIMS intercept date of SPG-IV than was assumed by Neymark et al., (2018).

However, this would still not fully account for the offset on the weighted mean at the ± 0.23 % 2s precision stated for Jian-1 (Neymark et al., 2018).

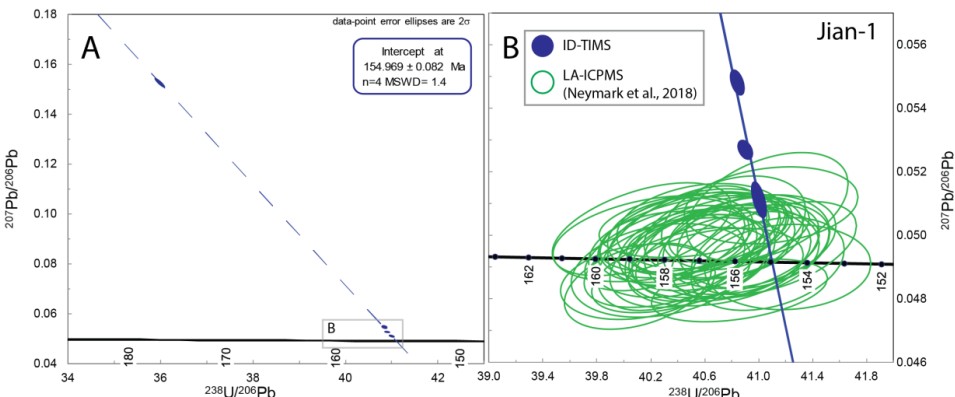

**Figure 4.** T-W plots of Jian-1. A) All ID-TIMS U-Pb data and isochron; B) ID-TIMS data excluding lowest Pb*/Pbc and isochron lower intercept compared with LA-ICP-MS data for Jian-1 Cassiterite (Neymark et al., 2018).

## 7. Evaluating the accuracy of cassiterite U-Pb dating: The geochronology of Cligga Head, SW England.

### 7.1 Geology and previous geochronology of Cligga head W-Sn deposit

The Cligga Head W-Sn greisen deposit is located at the central-northern periphery of the post-Variscan, Early Permian, SW England (Cornubian) Batholith, UK. Key features of the local geology (Fig. 5; Hall, 1971; Jackson et al., 1977; Moore and Jackson, 1977) are a porphyritic granite stock that intruded and locally thermally metamorphosed Devonian pelitic and psammitic meta-sediments at the contacts at about 1 kbar (Hall, 1971). The westerly extent of the granite has been eroded by the coast line. The main granite stock is cross-cut by a complex of sheeted quartz-muscovite (<1 % topaz and fluorite) greisen-bordered veins that predominantly contain quartz, in addition to tourmaline, chlorite, fluorite, cassiterite-stannite-arsenopyrite, wolframite and minor Cu sulfides. The cassiterite bearing greisen bordered veins extend across the contact into the meta-sediments.

Rhyolite porphyry dykes intruded ~150 m south of the stock. The absence of W-Sn mineralisation and greisenisation of the dykes, in addition to reported xenoliths of the granite greisen within the dykes





(Reid and Scrivenor, 1906), indicates they occurred after the W-Sn mineralisation, although they contain disseminated chalcopyrite and supergene derivatives (Moore and Jackson, 1977).

Previous geochronology of muscovite from the greisened zones indicated a date of ~280 Ma for the magmatic-hydrothermal system (Fig. 5). Two LA-ICP-MS U-Pb ages were presented (Moscati and

Neymark, 2019) for cassiterite from Cligga Head. A TW lower intercept age of 287.9 ± 2.6 Ma (n=59; MSWD = 2.2; initial $^{207}$Pb/$^{206}$Pb value of 0.790 ± 0.034) or an anchored isochron with a lower intercept of 289.2 ± 2.5 Ma (MSWD = 2.4). The weighted average Pbc corrected $^{206}$Pb/$^{238}$U date is 288.4 ± 3.0 Ma (n = 56, MSWD = 1.3). A second sample yielded lower intercept dates of 287.2 ± 4.8 Ma (n=40; MSWD = 2.1; initial $^{207}$Pb/$^{206}$Pb value of 0.839 ± 0.013 and a Pbc corrected weighted average $^{206}$Pb/$^{238}$U

date of 285.2 ± 4.2 Ma (n = 38, MSWD =1.3) (see Fig. 9).

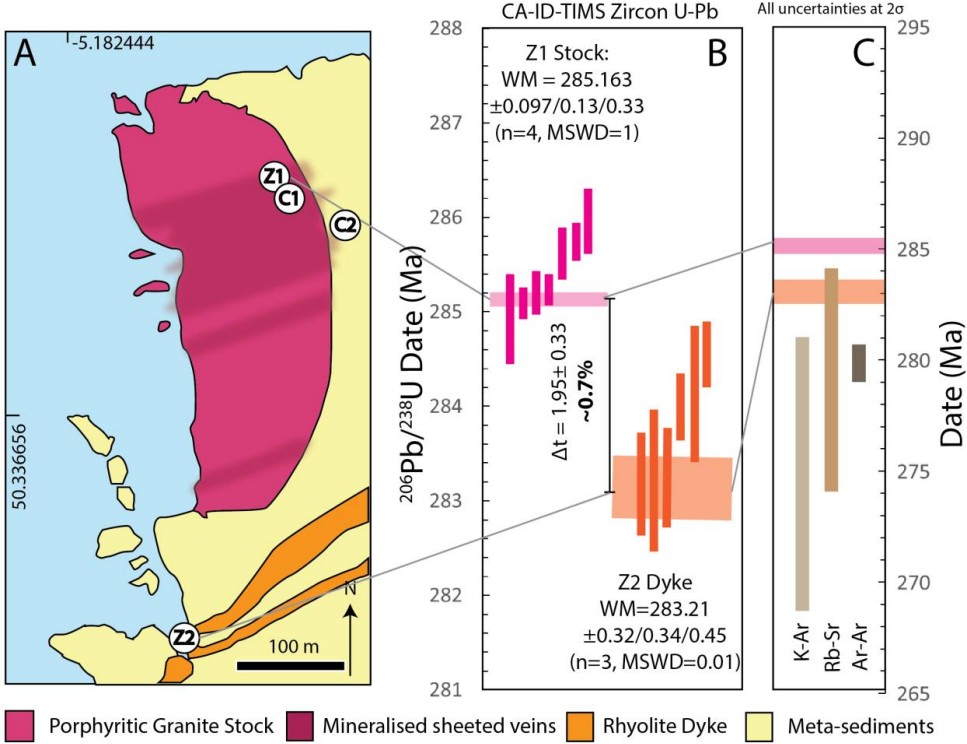

**Figure 5.** A) Geological map of Cligga Head (after Moore and Jackson, 1977) with localities of zircon samples (Z1 and 2) and cassiterite samples (C1 and C2) marked, pale blue signifies sea level; B) Zircon CA-ID-TIMS U-Pb dates and weighted mean ages for Cligga Head magmatism of the Porphyritic

granite stock (Z1) and  Rhyolite porphyry dykes. Dark shaded boxes represent single analyses of zircon

tips, pale boxes represent weighted mean dates. The intrusions define a period of ~0.7% of the absolute

age in which the cassiterite may have formed. Uncertainties presented as ±x/y/z where x = analytical

uncertainties only for comparison with cassiterite ID-TIMS U-Pb dates also using the ET535 tracer; y

= analytical and tracer calibration uncertainties; z = total uncertainty including $^{238}$U decay constant for

comparison with ages derived from other decay systems. C) Comparison of zircon CA-ID-TIMS U-Pb

weighted mean dates with previous geochronology for greisen system (Chen et al., 1993; Halliday,

1980).

**7.2 Samples and Zircon CA-ID-TIMS U-Pb geochronology methods**

Samples of the Granite Porphyry stock and the Rhyolite Porphyry dyke (Fig. 5) were analysed by zircon

chemical abrasion (CA)-ID-TIMS U-Pb in order to provide constraints on the timing of cassiterite

mineralisation. Zircon was analysed with the ET535 tracer and methods followed that of Tapster et al.

(2016). Data is corrected for initial $^{230}$Th disequilibrium using a Th/U(melt) of 1.43 derived from a

mean of granitic magmatism in the Cornubian Batholith (Simons et al., 2016). All data is contained

within the supplementary material.

**7.3 Zircon CA-ID-TIMS U-Pb results and the "age-window" for Cligga Head cassiterite**

The 10 analyses of zircon tip fragments from the host granite stock, yield a range in $^{206}$Pb/$^{238}$U CA-ID-

TIMS dates from ~290 Ma to ~285 Ma. The youngest four dates form a statistically acceptable weighted

mean of 285.163 ± 0.097 Ma (MSWD =1.01; n=4), which is interpreted as the emplacement age (Fig.

5).

The 8 analyses of zircon tip fragments from the porphyritic dyke which intrudes to the SE of the stock,

yield $^{206}$Pb/$^{238}$U dates from ~290 Ma to ~282 Ma. The youngest date, whilst overlapping with Concordia

within its relatively low single data point uncertainty demonstrates high discordancy (~11 %) and does

not form a single population weighted mean with any other dates, it is therefore attributed to residual

Pb loss and rejected from the age interpretation. The youngest dates that form a statistically acceptable

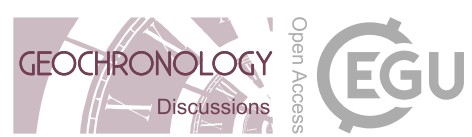

weighted mean date yield an age of 283.21 ± 0.032 Ma (MSWD = 0.01; n=3) taken as the timing of emplacement (Fig. 5).

These dates for the magmatism at Cligga Head provide independent minimum and maximum constraints on the absolute timing of the magmatic-hydrothermal greisen system in which the cassiterite formed when taken in the context of geological constraints. They define a possible window for the "true" age of cassiterite of 1.95 ± 0.33 Myrs, or about 0.7 % of the ~285 Ma absolute age.

**7.4 Cligga Head Cassiterite ID-TIMS U-Pb geochronology**

Cassiterite was sampled from two locations at Cligga Head (Fig. 5) from the first locality within the granite stock a single crystal (C1) was used within the decomposition experiments. In the first experiment a single part of the crystal was powdered into a single "parental" aliquot and divided into 8 approximately equal parts. We evaluated the effect of not leaching 4 of these cassiterite fractions in

29M HF relative to leaching, as described in the methods above and used in all of the other experiments (Fig. 6).

We then evaluated intra-crystal variation in U-Pb systematics by analysing an additional 3, spatially independent, parts of the same C1 crystal. To evaluate the inter-sample variation, fragments from 4 independent crystals were analysed from a second vein (C2) containing multiple cassiterite crystals that

was sampled from within the meta-sediments. Cassiterite ID-TIMS methods followed those described above (Sect. 5). All cassiterite U-Pb data is contained within the supplementary material.



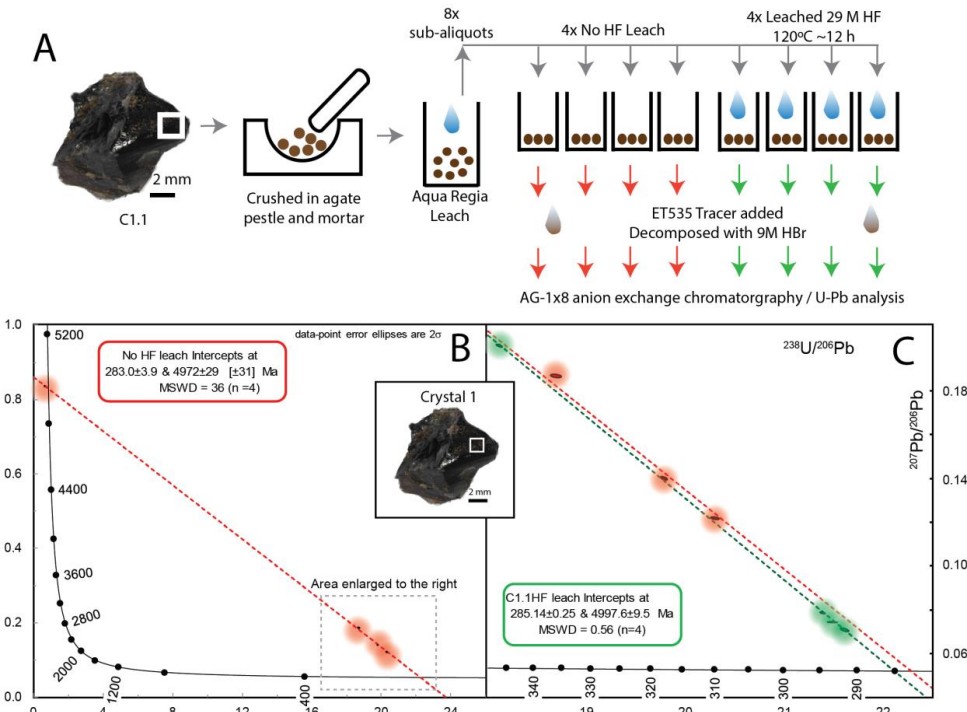

**Figure 6.** A) Schematic work flow of sampling and acid pre-treatment of C1.1 cassiterite. B) T-W plot and regression of non-HF leached sub-aliquots, individual data points are highlighted by a coloured halo due to their size at the scale of spread in data. C) T-W plot and regression of C1.1 HF leached sub-aliquots (green) compared to the data and regression of non-HF leached (Red) sub-aliquots. Data are highlighted by a coloured halo due to their size at the scale of spread in data.

**7.4.1 Comparison of HF leached and non-HF leached cassiterite from the same parental sample**

The sub-aliquots from the same parental powdered fragment of crystal (C1.1) that did not undergo HF leaching ranged from ~36 to ~55 pg of Pbc ($^{206}Pb/^{204}Pb$ = 108 to 207) and one aliquot yielded 7 ng of Pbc. Their regression line on the T-W plot forms a lower intercept of 283.0 ± 3.9 Ma with a statistically unacceptable MSWD of 36. The four sub-aliquots that were HF leached contained 6.7 to 47 pg of Pbc ($^{206}Pb/^{204}Pb$ = 97.7 to 691). Together they yield a lower intercept age of 285.13 ± 0.25 Ma with a statistically acceptable MSWD of 0.55 (n=4) and an upper intercept of 0.8760 ± 0.0058. None of the



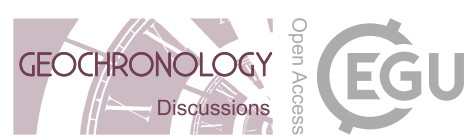

non-HF treated analyses form a single population (statistically acceptable MSWD) when combined with the regression line generated by the four HF treated sub-aliquots.

### 7.4.2 Intra-grain cassiterite U-Pb variability

The three spatially independent aliquots from the same crystal (C1.2-C1.4) did not yield a statistically

acceptable MSWD on their regression (lower intercept age = 284 ± 20 Ma; MSWD = 7.2; n=3). Only one of the three spatially distinct fragments of the same crystal fractions forms a single population with the HF leached C1.1 regression line, forming a lower intercept age of 285.19 ± 0.22 Ma (n = 5; MSWD = 0.73). All HF leached aliquots from the same crystal (C1) form a lower intercept age of 285.67 ± 0.72 Ma with a statistically unacceptable MSWD of 10.1 (n = 7). The total Pbc amounts for these analyses

ranges from ~3 pg to 8 pg and their $^{206}Pb/^{204}Pb$ range from ~400 to ~620.

Anchoring the lower intercept to the previously determined ~285 Ma lower intercept date of C1.1, and thus estimating the minimum possible $^{207}Pb/^{206}Pb$ upper intercept value due to the constraints offered by the granite age, yields $^{207}Pb/^{206}Pb$ upper intercept of 0.848, 0.807 and 0.783 (± ~2 %), lower than the previously determined upper intercept of ~0.876 for the same crystal.

### 7.4.3 Inter-vein and –grain variation in U-Pb systematics

No analyses derived from individual crystals of the second hand sample (C2) form a single population when integrated with data that formed the isochron of C1.1. Neither do the four analyses form a single population between themselves, yielding a lower intercept of 286.2 ± 5.0 Ma with a statistically unacceptable MSWD = 48 (n = 4) and $^{207}Pb/^{206}Pb$ upper intercept of 0.72 when freely regressed and

0.66 (± 5 %) when anchored to the C1.1 lower intercept date and maximum possible age derived from the granite of ~285.1 Ma. The total Pbc amounts for these analyses ranges from ~4 pg to 17 pg and their $^{206}Pb/^{204}Pb$ range from ~430 to ~1300.

Again, to estimate the maximum initial $^{207}Pb/^{206}Pb$ by anchoring the lower intercept of each date the maximum possible age intercept defined by the granite yields upper intercept initial $^{207}Pb/^{206}Pb$ for

each data point of 0.695, 0.643, 0.595 and 0.561 (± 0.7-2.3 % 2s), significantly lower than the value derived for C1.1 (~0.876).

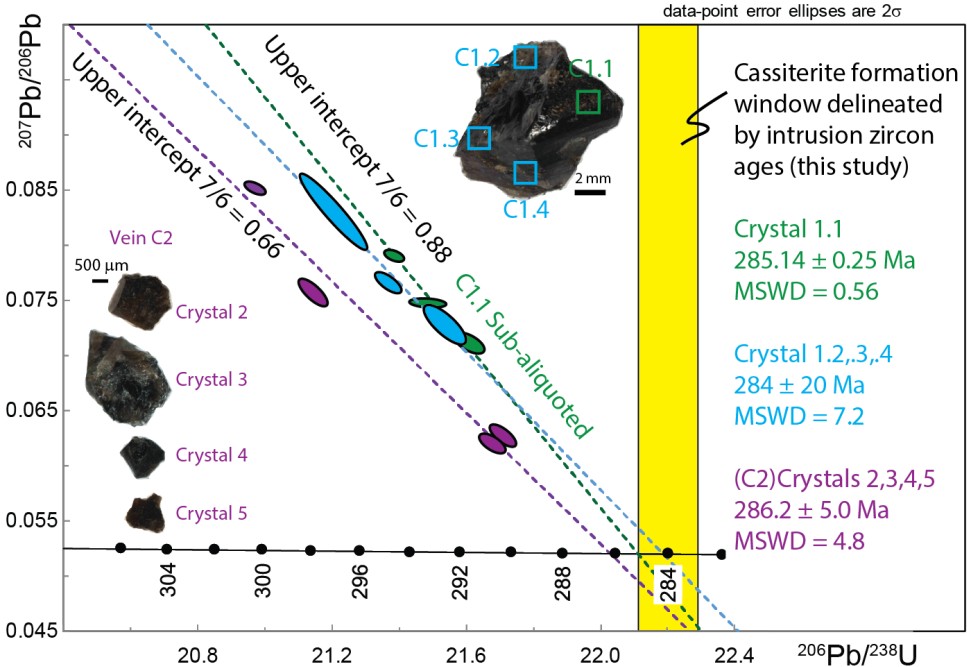

**Figure 7.** T-W plot and regression lines for HF leached cassiterite from Cligga Head relative to possible age defined by zircon (Fig. 5). Regressions are for: C1.1 sub-aliquots (n=4) as presented in Fig. 6; Spatially independent samples of the same crystal (n=3); and independent crystals from a different vein (C2) (n-4), note upper intercept value is for regression anchored to maximum possible lower intercept as defined by the granite, it therefore represents a maximum upper intercept value. Only the sub-aliquots of C1.1 yield a regression without over dispersion. See text for further discussion.

## 8. Discussion

This cassiterite U-Pb dataset, combined with prior work (e.g. Li et al., 2016; Moscati and Neymark, 2019; Neymark et al., 2018; Zhang et al., 2017) highlights the potential for U-Pb dating of cassiterite using both microbeam and isotope dilution methods. Such datasets also reveal some complexities and issues that need to be considered when deriving meaningful ages from cassiterite dates.

### 8.1 Assessing U-Pb systematics of (Cligga Head) cassiterite

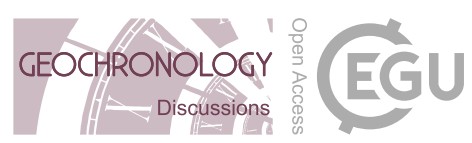

The range of ID-TIMS experiments described above permit us to evaluate the U-Pb systematics of cassiterite at high-precision and within the context of independent geochronological constraints derived from zircon CA-ID-TIMS dating within a well constrained geological model. There are three key

features of the data set:

**1.** The lower intercept date formed by a single population of HF leached cassiterite sub-aliquots of the same parental material (C1.1), indicating binary mixing line between an initial Pb and a radiogenic Pb source. The lower intercept yields a precise date that is consistent with the indirect age constraints on cassiterite U/Pb*.  The effects of any initial isotopic disequilibrium have to be considered when deriving

a sample age and uncertainty (see below).

**2.** The over-dispersion of non-HF leached cassiterite compared to HF leached C1.1 cassiterite analyses (Fig. 6) indicating non-binary mixing between initial and radiogenic Pb. The non-HF leached experiments contained a greater contribution of Pbc, with the high-MSWD indicating different sources/isotopic compositions of Pbc that varied in magnitude of contribution between sub-aliquots.

Both sets of experiments (Fig. 6) were treated with aqua regia prior to analysis as with previous cassiterite ID-TIMS studies (Gulson and Jones, 1992). These results therefore indicate that leaching with concentrated HF is an effective means to remove the Pbc contained within inclusions within the cassiterite that are exposed to the surface during crushing in a pestle and mortar. The results also indicate HF leaching is effective in removing Pbc bearing contaminants introduced by crushing in pestle and

mortar, as proposed as a potential issue by Neymark et al. (2018). The acid resistant nature of cassiterite and ability of HF to dissolve silicate minerals without partially dissolving the cassiterite and potentially leading to incongruous removal of Pb or U, makes this an important step to remove non-lattice bound Pbc prior to decomposition by HBr and improve the accuracy of resulting ID-TIMS U-Pb dates.

**3.** The differences between spatially independent samples from within and between crystals, and

between cassiterite from different veins (Fig.7), indicating localised controls on the U-Pb systematics of cassiterite. Only one spatially independent fragment from the same crystal of cassiterite forms a single population with the isochron of HF leached C1.1 described above. A dataset based upon aliquots of the same crystals (C1.2, C1.3, C1.4) yield a similar age to the C1.1 isochron but is over-dispersed



and the lower intercept is less precise. A similar cassiterite U-Pb ID-TIMS dataset from a different vein

(C2) is also over-dispersed and plots distinctly to the left of the C1 mixing array between the constrained

crystallisation age and initial Pbc $^{207}$Pb/$^{206}$Pb compositions.

Open system behaviour with respect to Pb (Pb-loss) would shift data points to the right (higher U/Pb)

of the C1.1 (or similar) isochron. Estimated uranium concentrations (see Cassiterite ID-TIMS U-Pb

data table in supplementary materials) are similar between cassiterite from the two veins and as such

there is no apparent reason why uranium mobility would be manifest to a greater extent between

different veins. We consider that the most likely explanation is that the over-dispersion results from

variation in the initial Pbc of the cassiterite to more "radiogenic" initial $^{207}$Pb/$^{206}$Pb values than ~0.88

as defined by the C1.1 isochron, potentially as low as ~0.56.

The majority of documented $^{207}$Pb/$^{206}$Pb from feldspars in granites and fluid inclusions in the Cornubian

Batholith are ~0.8-0.9 (Hampton and Taylor, 1983; Wayne et al., 1996), with only one magmatic

feldspar analysis yielding a $^{207}$Pb/$^{206}$Pb of  ~0.55 (Hampton and Taylor 1983). It is unlikely that these

low values reflect the primary input signature from the magmatic volatiles. Initial Pb with $^{207}$Pb/$^{206}$Pb

this low are uncommon but are present in a global array of vein carbonates (Roberts et al., 2019). When

the nature of cassiterite-bearing greisen systems, such as Cligga Head, is taken into account, the ability

to generate atypical Pbc isotopic compositions is less surprising. Geisenisation at the margins of the

cassiterite-bearing veins represents extensive, destructive, fluid-rock interaction and elemental

exchange during alteration of host rocks by high temperature ~450-300°C, highly acidic, fluorine-rich

(HF) fluids (Burt, 1981; Codeço et al., 2017; Lecumberri-Sanchez et al., 2017). As at Cligga Head, W-

Sn greisen bordered vein magmatic-hydrothermal deposits are commonly hosted by, or are proximal to

thick continentally derived sedimentary packages (Lecumberri-Sanchez et al., 2017) that likely

represent the source of Sn enrichment prior to magma genesis (Romer and Kroner, 2016). These

sediments will contain, older U- and Pb*- rich continental mineral detritus (e.g. zircon, monazite) that

can be extensively leached by the hot, F-rich, acidic ore-forming fluid generating significant spatial and

temporal variations in the Pbc isotopic composition incorporated into the cassiterite lattice over the

lifetime of single crystal precipitation, and the magmatic hydrothermal system as a whole. It is perhaps



significant that the cassiterite that appears to contain the lowest initial Pbc component originates from

a vein bordered by greisened meta-sediment rather than hosted within the granitic stock.

### 8.2 Cassiterite U-Pb ages and effects of initial U-Th disequilibrium.

The high-precision U-Pb lower intercept cassiterite date (C1.1; Figs. 6 and 7) comes with the caveat

that it is not corrected for potential initial isotopic disequilibrium effects within the $^{238}$U decay chain.

In zircon geochronology a deficit in $^{206}$Pb due to $^{230}$Th disequilibrium (Schärer, 1984) can be reasonably

corrected using a reasonable assumption of the mineral/melt partition coefficient or an estimate of melt

Th/U, and an estimate of mineral Th/U, typically based on assumed $^{208}$Pb/$^{232}$Th-$^{206}$Pb-$^{238}$U concordance

for zircon ID-TIMS data. Cassiterite typically has extremely low $^{232}$Th relative to U (Th/U = <10$^{-4}$

Neymark et al., 2018). This geochemical feature may be derived from preferential partitioning of U into

the ore fluids, and/or preferential partitioning of U upon precipitation of cassiterite leading to a deviation

from secular equilibrium. Both processes will require a significant correction for a deficiency in $^{206}$Pb

due to a low $^{230}$Th/$^{238}$U activity in the initial cassiterite. Alternatively this geochemical feature could

simply be explained by processes that do not preclude secular equilibrium with regards to $^{230}$Th upon

cassiterite crystallisation. These include low $^{232}$Th/$^{238}$U magmatic ore fluid sources, or the contribution

of localised U-rich components to fluids that are in secular equilibrium with regards to $^{230}$Th and $^{238}$U

during alteration around veins. In these cases the $^{232}$Th-$^{208}$Pb system would not provide a realistic proxy

for initial $^{230}$Th disequilibrium in cassiterite.

It must also be considered that the hydrothermal fluids, and therefore the initial cassiterite may not have

been in secular equilibrium for $^{238}$U-$^{234}$U.   A relative excess of $^{234}$U, or a $^{234}$U/$^{238}$U activity ratio >1, at

the time of cassiterite precipitation will result in an excess of $^{206}$Pb/$^{238}$U, an older measured date relative

to the true age. These effects therefore counteract effects of $^{206}$Pb deficiency due to $^{230}$Th disequilibrium.

Mixing with meteoric fluids appears to be an important process in cassiterite formation (Fekete et al.,

2016) and as highlighted by the compilation of Roberts et al. (2019) the $^{234}$U/$^{238}$U activity of crustal

fluids can be in excess of 1, (shallow ground waters median value of 2.25; hydrothermal fluids median

of 1.41).



Estimates of the effects of initial disequilibrium (Fig. 8) indicates that corrections for the exclusion (or

absence) of initial $^{230}$Th, potentially inferred by the characteristic low $^{232}$Th/U in cassiterite, produces a

maximum possible date (285.23 Ma) that is still within uncertainty of the maximum age permitted by

the granite host.  Effects of excess initial $^{234}$U indicate that an activity ratio of 1.5 produces dates ~200

kyrs younger (284.92 Ma) than that of secular equilibrium, yet also lies within the analytical

uncertainties of the host granite.  In the more extreme scenario of a $^{234}$U/$^{238}$U activity ratio of 2.5, this

reduces to 284.55 Ma and would be significantly younger.

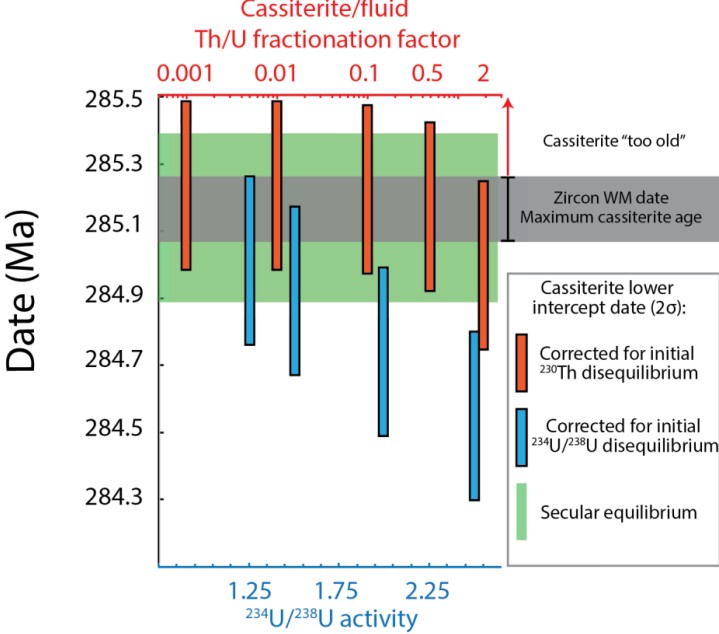


**Figure 8.** Effects of corrections for initial intermediate daughter disequilibrium on the lower intercept
date of C1.1 regression over geologically feasible parameters. Note that corrections are treated
independently and that combinations of initial disequilibrium effects could be present. The maximum
possible age of cassiterite is defined by the zircon age and uncertainty of the Porphyritic Granite stock
that hosts the cassiterite bearing veins.  See text for discussion.

Not accounting for the  uncertainty that arises from the potential effects of disequilibrium, the lower

intercept of the four point isochron C1.1 yields a precision of ± 250 kyrs or 0.088 % (2σ, analytical





uncertainties only,) and an agreement with the possible window for hydrothermal activity defined by

the CA-ID-TIMS zircon dates of the intrusions (~0.7 %). Moreover it also demonstrates a clear temporal

association with the granitic stock that hosts the Cligga Head deposit (the cassiterite date is 0.021 ±

0.268 Myr younger than the zircon date for the granitic stock) (Fig. 8, 9). High precision constraints on

the durations of relatively simple ore-forming magmatic-hydrothermal systems spatially associated

with single intrusions indicates timeframes < 10s to 100s kyrs  (Li et al., 2017; Tapster et al., 2016). It

is therefore feasible that the emplacement of the host granite stock, cooling to the 400°C-350°C

temperature interval of cassiterite precipitation (Smith et al., 1996), and transfer of ore forming volatiles

from depth, all occurred within the timeframes defined by the <0.1 % uncertainty of the zircon and

cassiterite data presented here. However, considering the potential uncertainty due to initial U-Th

disequilibrium effects over geologically reasonable assumptions the cassiterite uncertainty could be

expanded by ca. -200 kyrs and would be asymmetric (Fig. 8).  At present the poor understanding of

isotopic disequilibrium effects due to elemental partitioning into ore fluids, and cassiterite represent a

limitation for the accurate interpretation of absolute ages of cassiterite beyond the  <~500 kyr timescale.

### 8.3 Implications and strategies for LA-ICP-MS U-Pb cassiterite geochronology

In this study we have generated ID-TIMS U-Pb data on cassiterite that have undergone full

decomposition in a single stage.  These include samples that have previously been used for LA-ICP-

MS studies using an approach to U/Pb normalisation using an ad hoc cassiterite reference material with

an inferred U/Pb age.  The comparison of the ID-TIMS and LA-ICP-MS U-Pb data provide a direct

means to assess the accuracy of the microbeam dates (Moscati and Neymark, 2019; Neymark et al.,

2018), which show agreement at the ca. 1 % level and is therefore comparable to the quoted levels of

accuracy and similar to the accuracy for U-Pb (zircon) studies (Horstwood et al., 2016). Additional

characterisation and over a wider range of cassiterite materials by the ID-TIMS methods described here

will establish a focal point for the refinement of cassiterite measurement uncertainties by LA-ICP-MS.

The ID-TIMS U-Pb data from the Cligga Head cassiterite strongly suggests that the initial Pb isotopic

composition varies and that a simple binary mixture between a radiogenic and initial Pb may not be



expected within and between crystals. A high proportion of LA-ICP-MS U-Pb cassiterite datasets record over-dispersion (e.g. ~2/3 of samples, typically analysed from single crystals, reported by Neymark et al., 2018) suggesting that these are also not strict binary mixtures. We postulate that some of the over-dispersion can be attributed variable initial Pb isotopic compositions within cassiterite that

is inherited from the hydrothermal systems. This complexity in the U-Pb systematics represents a potential limitation on the accuracy of the resulting age interpretation of cassiterite. This limitation is interdependent with the single data point precision of the U-Pb analyses and the ability to resolve different populations of the Pbc mixing endmember that will bias the regression of a data set away from the accurate lower intercept.

We explore how the U-Pb data can be leveraged by variable Pbc by correcting the Cligga Head data set by using a widely employed strategy for minerals that accommodate Pb into their lattice upon formation, of correcting the component of initial Pbc using a model Pbc value at ~285 Ma (Stacey and Kramers, 1975). Figure 9 demonstrates the resulting Pbc corrected $^{206}$Pb/$^{238}$U dates range from being in good agreement with the host granite age to being 'too old' by ca. 4 Myrs (~1.5 %). Without the maximum

constraint from the age of the granite, these 'model' U-Pb cassiterite dates alone could be misinterpreted as a protracted episode of mineralisation. Comparable shifts, both in magnitude and direction, can be observed within one of the two samples from the same deposit that were analysed by LA-ICP-MS U-Pb (Moscati and Neymark, 2019; Fig. 9).

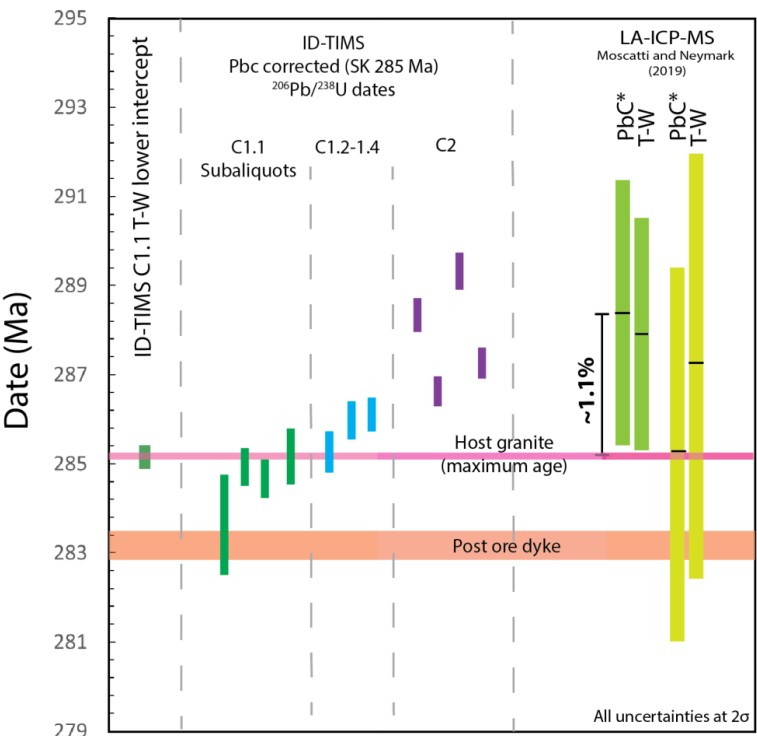

**Figure 9.** The effects of using a correction for Pbc on the [206]Pb/[238]U dates of single analyses of cassiterite, in this case a model Pbc value at 285 Ma (Stacey and Kramers, 1975) was used. Data are shown relative to the freely regressed ID-TIMS lower intercept of C1.1 that shows excellent agreement with the host granite age. Pbc corrected dates range from being in good agreement with the age constraints to being offset by ~4 Myrs relative to the maximum possible age of cassiterite defined by zircon dates for the granite. The samples analysed in a previous LA-ICP-MS study from Cligga Head (Moscati and Neymark, 2019) illustrate similar magnitudes and direction of offset can be observed in both the T-W regressions. Despite the agreement of the Pbc corrected (Pbc*) weighted mean [206]Pb/[238]U date 290 Ma (Stacey and Kramers, 1975) of one sample, this approach to data interpretation is invalid, see text for discussion.

To further illustrate this effect, and the impact of larger single data point uncertainties on resolving dispersion, an isochron based upon all Cligga Head cassiterite ID-TIMS U-Pb analyses but with uncertainties expanded to ± 1 % (2σ) precision, shows no dispersion (MSWD = 1.2 and statistically





acceptable), however the resulting age is biased too old (lower intercept at $286.80 \pm 0.95$ Ma; MSWD

$= 1.2$, n = 12) for the known maximum possible age that is defined by the granite. The approach of

taking weighted means of common Pb corrected data is strongly advised against as it diminishes the

evaluation of dispersion and ignores the uncertainty of $^{207}Pb/^{206}Pb$ measurement (see Horstwood et al.,

(2015) for community discussion), but as an illustration of the effects of variable Pbc composition, for

the same data set with uncertainties expanded to 1% also lies outside of uncertainty of the maximum

possible age weighted mean date of Pbc corrected $^{206}Pb/^{238}U$ dates $286.23 \pm 0.83$ Ma (MSWD = 1.3; n

= 12). Any bias resulting from Pbc variation would be in addition to greater scatter on individual data

points resulting from lower precision measurements. This reinforces the limitations on the accuracy of

the interpreted ages when derived from high-n but lower precision data sets when there are potential

variations in the initial Pbc isotopic composition, because they can be masked by the single data point

uncertainty yet impact on the accuracy of the age. This should be taken into account, in addition to the

limits of systematic uncertainties defined by validation RMs, within future work of both cassiterite and

other minerals with initial Pb.

We note that the reference values of any cassiterite material used as an RM should not be corrected for

Pbc, or any initial isotopic disequilibrium effects, rather they should reflect the isotopic values of

material ablated (Horstwood et al., 2016), as with the data presented in Figs. 3, 4, 6 and 7. As HF

leaching of cassiterite targets inclusions and is unlikely to affect lattice bound Pb we suggest this will

not modify the isotope ratios as chemical abrasion does for zircon RMs.



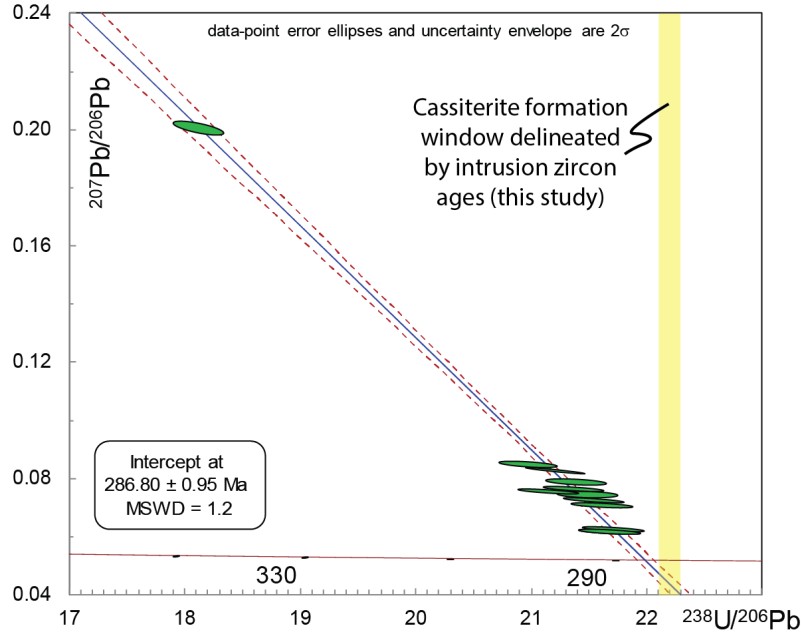

**Figure 10.** Illustration of how variable Pbc compositions in cassiterite form a "single population", yet

with an inaccurate T-W lower intercept within lower single data point precision data sets. The data are

all HF leached ID-TIMS U-Pb data from Cligga Head with artificially expanded single data point

uncertainties of 1% (2s) for both $^{238}U/^{206}Pb$ and $^{207}Pb/^{206}Pb$. This level of uncertainty is therefore a

reasonable illustration of the upper range of analytical precision that may be achieved through

microbeam techniques in relatively low U concentration materials such as cassiterite. Note the data with

artificial uncertainties does not show excess dispersion (statistically acceptable MSWD) but produces

a relatively precise lower intercept (0.33 %) outside of uncertainty and older than the maximum

permitted age of cassiterite delineated by the granite (Fig. 5). This does not factor in any additional

systematic or analytical inaccuracy from lower precision methods.


## 9. Conclusions

The absolute dating of Cassiterite using the U-Pb decay system offers the potential to link the

hydrothermal processes responsible for Sn mineralisation to the regional and local (deposit) scale

magmatic-hydrothermal systems that may or may not be related to their genesis. Recent studies (e.g.

Liu et al., 2007; Moscati and Neymark, 2019; Neymark et al., 2018; Yuan et al., 2011, 2008; Zhang et



al., 2017; and others) have demonstrated the potential of cassiterite for U-Pb geochronology using microbeam methods (LA-ICP-MS) applied to the mineral from a wide range of deposits. The lack of materials that have characterised U-Pb compositions via total dissolution isotope dilution methods, and the difficulties in obtaining these, has led to the development of calibrations using inferred ages for ad

hoc reference materials and impedes the verification of U-Pb datasets generated.

We have demonstrated that useful amounts of cassiterite can be fully decomposed under closed system conditions, using readily obtainable and low Pbc blank HBr acid, over timescales and using apparatus similar to that used in zircon ID-TIMS work. Increasing the surface area/volume ratio (e.g., using a pestle and mortar) is in fact an advantage when utilised with an HF leaching step, as it not only expedites

cassiterite decomposition, but will also expose Pbc-bearing inclusions that can be leached and rinsed away, along with any material abraded during crushing in a pestle and mortar. The notorious acid resistance of cassiterite here works to our advantage as HF leaching leaves the lattice bound U-Pb systematics intact. The methodology presented here indicates that Pb can be isolated using a simple modification of AG-1x8 resin HCl based anion exchange chromatography typically used for zircon,

with no detectable penalties for ionisation by TIMS, and a further AG-1x8 HCl and $HNO_3$ based scheme is effective for the U elution.

The U-Pb ID-TIMS data presented for a range of materials between ~150 to ~1600 Ma, demonstrate that cassiterite ID-TIMS U-Pb geochronology can potentially generate U-Pb lower intercept dates at ± <0.1 % precision, from a relatively low number of analyses. Using the classic example of the W-Sn

Greisen deposit at Cligga Head, UK, we demonstrate that the implementation of "internal isochrones" from sub-aliquots of powdered fragments cam derive a U-Pb ID-TIMS cassiterite age interpretation consistent with an independent high-precision zircon CA-ID-TIMS U-Pb constraint on the age of cassiterite. These direct constraints show that the hydrothermal system was relatively short lived illustrating the potential of this method to define causative magmatic events. However, unknowns

regarding initial isotopic disequilibrium of cassiterite probably limit the true uncertainty on the age to ~400 kyrs.

Comparison of ID-TIMS and previously reported LA-ICP-MS U-Pb datasets for the three samples indicate that the latter are accurate at the ~1 % (2σ) level, especially when taking into account the ID-TIMS date that was ~0.4 % younger than the inferred age for the SPG-IV cassiterite used to normalise

matrix effects in the Neymark et al. (2018) study. This direct U/Pb determination can be used to refine LA-ICP-MS data that are normalised to this ad hoc reference material, further improving the underpinning of cassiterite U-Pb via LA-ICP-MS. Further total digestion ID-TIMS U-Pb analyses of cassiterite will further improve the method, hopefully providing additional reference materials for calibration and verification.

The high precision data presented here indicates that intra- and inter- grain variations occur in the isotopic composition of Pbc present within the cassiterite. The nature of greisen Sn hydrothermal systems and their association with continentally derived sedimentary packages, may be conducive to localised radiogenic Pbc compositions. The potential for these variations in Pbc compositions places limitations on the interpretive power of data sets as a function of single data point precision. A cautious

approach to the age interpretation of lower precision micro-beam data sets, beyond the issues of analytical precision, is advised.

## 10. Data availability

Data to reproduce all plots presented here is provided within the supplementary material excel workbook.

## 11. Author contributions

ST and JB designed the experiments that were carried out by JB with contributions from ST. ST interpreted the data and prepared the manuscript with contribution from JB.

## 12. Competing interests

The authors declare no competing interests.

## 12. Acknowledgements



The authors are extremely thankful for the provision of cassiterite materials by Richard Taylor; Richard Shaw; Leonid Neymark (USGS). We thank Jeremy Rushton and Gren Turner at the BGS

SEM facility. The authors thank Dan Condon and Matt Horstwood for discussion and comments on this manuscript.

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
