# Peer review of "High-precision ID-TIMS Cassiterite U-Pb systematics using a low-contamination hydrothermal decomposition: implications for LA-ICP-MS and ore deposit geochronology"

_Geochronology, 2019_

## Short Comment (SC1) · 13 Feb 2020

This is a nice study with useful results. Just for information, the following paper was accepted for publication in Chemical Geology on 12 February 2020:

"A new method for U-Pb geochronology of cassiterite by ID-TIMS applied to the Mole Granite polymetallic system, eastern Australia." by P. Carr, S. Zink, V. Bennett, M. Norman, Y. Amelin, and P. Blevin.

[Figure]

As such, this Geochronology manuscript should be revised to reflect this prior publication with appropriate citation (e.g., line 15, "... due to the extreme resistance of cassiterite to most forms of acid digestion, to date, there has been no published method permitting the complete, closed system decomposition of cassiterite under conditions where the basic necessities of measurement by isotope dilution can be met, leading to a paucity of reference, and validation materials", and line 33: "this method can, for the first time, be used to properly characterise suitable reference materials for micro-beam cassiterite U-Pb analyses".

The chemical separations used by the two studies are somewhat different so a brief comparison of the methods and results might be appropriate in the revised version as well.

Thank you to the authors for presenting a clear and well described study.

M. Norman, RSES-ANU 13 Feb 2020

---

## Referee Comment (RC1) · Corey Wall (Referee) · 18 Feb 2020

Manuscript number: gchron-2019-22 Title: High-precision ID-TIMS Cassiterite U-Pb systematics using a low-contamination hydrothermal decomposition: implications for LA-ICP-MS and ore deposit geochronology

The manuscript provides a clear, detailed, and well thought out study to determine a novel method that allows the complete digestion of cassiterite and its utilization in ore

deposit geochronology. It is relevant to a broad community and this is the appropriate platform for publication. Overall, the science of this manuscript is of the highest quality and with minor revisions is suitable for publication. The authors do a great job discussing the current issues with U-Pb cassiterite geochronology and laying out a laboratory procedure for digestion, chemistry, etc. The updated ages for cassiterite reference materials and insights into the U-Pb systematics of cassiterite, this study will prove very useful for future in-situ studies. With a few changes, this manuscript should be ready for publication.

When I began reading this manuscript I was very excited to see the results and interpretation of the U-Pb cassiterite data; however, by the end I was left a little confused by the impact the results have on the overall interpretation of the geologic problem. For example, based on the geological relationships of the Cligga Head W-Sn deposit, we know that Sn ore formation occurred between the granite stock (285.163 +/- 0.097 Ma) and rhyolite porphyry dykes (283.21 +/- 0.32 Ma). Although the cassiterite age falls in this "age window", there are too many variables associated with this age that it becomes difficult to put geologic implications or a direct process (i.e. cooling of the granitic stock and precipitation of cassiterite). We know roughly what age the cassiterite has to be, but it would be nice to directly link it to a geologic process, and unfortunately it seems impossible at this time. Based on this, are there any other phases you could determine the Pb isotopic composition of? What about the inclusions in the cassiterite that you dissolve during the HF leaching step? I think there is an interesting study looking at the Pb isotopic composition in these low T hydrothermal systems.

Based on the heterogeneity in Pbc seen in the Cligga Head cassiterite, do the authors believe that this could be the case for the two RM analyzed in this study? It is noted on page 15 for the Jian-1 cassiterite that domains with lower Pb*/Pbc were sampled, could there also be zones with variable Pbc isotopic composition? Thinking about in-situ work for this.

I would like to see a final figure that describes the finalized workflow the authors have

determined to be the most effective way to dissolve cassiterite so that labs looking to adopt this method have something to follow. A modification of Figure 6A) would be nice to illustrate the full procedure. Do the authors think a pestle and mortar approach to cassiterite is a good option, or should individual grains or fragments of grains be analyzed? This is something that should be addressed in the conclusion.

The conclusions section needs to be reworked. The authors did a great job characterizing the issues with U-Pb cassiterite geochronology and make great strides forward in this field, but the last paragraph and concluding sentences of paragraphs leaves me curious of what the actual outcome of this manuscript is. The last paragraph in the conclusions should be removed entirely and replaced. The end of the abstract should be revisited by the authors to aid in writing a satisfactory conclusions section.

Line by line comments-

Line 12 - Change "the mineral" to cassiterite, add comma after "common Pb" Line 15 - delete "to date," Line 17 - delete "," after reference Line 26 - should be "demonstrates" Line 32 - delete "However," Line 39 - add some references at the end of the first paragraph Line 46 - delete "Yet" Line 77 - delete "However, it" and replace with "This" Line 90 - add "in-situ" between resulting and ages Line 91 - This is a run on sentence, consider breaking these thoughts into two sentences Line 104 - Awkward sentence, needs reworking Line 107 - Again, a run on sentence, consider breaking these thoughts into two sentences Line 114 - delete "that" after 2) Line 115 - delete "That" after 3) Line 137 - delete "and at the time of writing was" and replace with is Line 138 - comma after "Pb", delete "was" and replace "Likewise" with Similarly Line 141 - Delete "Furthermore" Line 150 - Give range for what is typically achieved for modern CA-ID-TIMS methods Line 152 - changed "indicate" to indicated Line 158 - delete "the study of" Line 159 - add comma after (1992) Line 164 - add reference Scoates and Wall, 2015. Geochronology of Layered Intrusions Line 167 - Delete "yet" before "been" Line 183 - delete comma after Br2 Line 221 - delete "before the HBr is lost and becomes ineffective" Line 243 - The authors mention an HF leaching step; however, this leaching step is not described

anywhere in this preparation step. Is it the 120C overnight in 29M HF? If it is then explicitly say that and if it was in Parr vessels or on the hotplate. This is a paper outlining the proper way to digest cassiterite, be very specific about these details Line 255 - do you worry about the longevity of the PFA microcapsules? If there is degradation of the capsules at slightly higher temperatures do you worry about exposure to HBr over longer periods of use? These capsules can be very time consuming to clean before initial use Line 267 - Did the authors consider an HBr based chemistry for Pb? This would remove the HCl conversion step and reduce the laboratory blank slightly. Line 291 - What decay constant and isotopic compositions for U were used? Since this is a paper about method development and reference materials, this information is critical. Line 420 - Again, this was not described thoroughly in the methods section Line 424 - Were these HF leached as well? This treatment that the various cassiterite grains received are not well documented here, clarification is needed Line 436 - do the authors think this 7 ng blank for the one analysis is an inclusion? Line 439 - delete "Together they yield" and replace with "and yielded" Line 590 - a poor way to end a discussion session, try rewording this so it does not have such a negative tone to it. Maybe delete this sentence and end with the previous sentence Line 606 - this is a very troublesome sentence, how do the authors suggest combating this variable Pbc mixing or is it a problem that can be solved? If the latter is the case, then it severely hinders the interpretation of cassiterite U-Pb ages. Line 614 - Again, any suggestions on how to correct for this? Line 655 - delete "As" Line 696 - replace "cam" with "can"

Does the paper address relevant scientific questions within the scope of GChron? Yes Does the paper present novel concepts, ideas, tools, or data? Yes Are substantial conclusions reached? Yes Are the scientific methods and assumptions valid and clearly outlined? Yes, with modifications Are the results sufficient to support the interpretations and conclusions? Yes Is the description of experiments and calculations sufficiently complete and precise to allow their reproduction by fellow scientists (traceability of results)? Yes, with modifications Do the authors give proper credit to related work and clearly indicate their own new/original contribution? Yes Does the title clearly reflect the

contents of the paper? Yes Does the abstract provide a concise and complete summary? Yes Is the overall presentation well structured and clear? Yes Is the language fluent and precise? Yes Are mathematical formulae, symbols, abbreviations, and units correctly defined and used? Yes Should any parts of the paper (text, formulae, figures, tables) be clarified, reduced, combined, or eliminated? Yes (stated above) Are the number and quality of references appropriate? Yes Is the amount and quality of supplementary material appropriate? Yes

---

## Referee Comment (RC2) · Gavin Piccione (Referee) · 13 Apr 2020

Summary and general comments: Tapster and Bright present a novel, low-blank laboratory procedure for U-Pb ID TIMS measurements. The authors successfully apply their method to produce U-Pb dates for three cassiterite samples that have previously been dated by LA ICP-MS (Neymark, et al., 2018; Moscati, et al., 2019), and thus provide a comparison between their method and levels of precision reported in the

most recent studies using micro-beam measurements. They also utilize the increased levels of analytical precision provided by their method to assess the reproducibility of cassiterite U-Pb dates on both a single-grain and multi-grain population.

In this manuscript, Tapster and Bright provide U-Pb ages for 2 potential cassiterite micro-beam reference standards. Their datasets, while having a considerably smaller sample size than previous LA ICP-MS datasets, benefit from much lower uncertainties on each single data point and therefore provide more tightly constrained ages than those previously published. They measure U-Pb cassiterite ages for the Cligga Head W-Sn greisen deposit and show that cassiterite mineralization was coeval with magma emplacement in the area. Results Cligga Head samples also show that the initial Pb composition of cassiterite can be heterogenous on a single grain scale, which can result in biased U-Pb micro-beam ages.

Overall, I think the data provided in this paper are of high quality, the experiments, results and conclusions are sound, and the authors successfully show the promise of their new method. The authors do a nice job of identifying challenges to U-Pb cassiterite geochronology that should be considered in future studies. This contribution is particularly timely, as there is growing global interest in high-precision analyses of ore-bearing mineral deposits, with many recent publications utilizing U-Pb cassiterite dating. It is my opinion that this paper fits well within the scope of Geochronology and I recommend it for publication following minor revisions.

My one general critique of the manuscript is that the text, while well structured, could benefit from revision to present the main points more clearly, and to fix several instances of typos and run-on sentences. I also suggest that the authors make sure that they call out all of their figures in the text.

Specific Comments: Methods: 1) To justify that they achieved full decomposition they use SEM and EDS scans of post HBr dissolution solids and show that there are no Sn-oxides left. Why don't they include these data? Given that full decomposition is of

fundamental importance to this paper, I suggest that you include these data if not in the main text, in the supplement.

2) I have several questions regarding the HF leaching procedure: How long do you leach in HF (you might say this on line 242 but it's not clear that you mean that you refluxed samples in HF overnight)? How much HF do you use? Do you rinse after HF leaching? Do they lose any material after leaching? I think the paper could benefit from a clearer description of the HF leaching process.

3) You mention that the HF procedure effectively removes Pbc from inclusions that are exposed during powdering with mortar and pestle. Do you feel that this is 100% affective? Although it appears to be on C1.1, could micro-inclusions or variable levels of leaching be responsible for the differing Pbc composition in C1.1-4?

4) It is likely that variation in initial Pb composition in a single grain of cassiterite would be unavoidable, but could you decrease the risk of sub-grain scale heterogeneity (like that in C1) by imaging (SEM or CL) the grains before ID TIMS analyses? I acknowledge that most inclusions would be much smaller than the ID TIMS sampling size, but large-scale features like secondary growth zones, imbedded grains, or secondary mineral growth may be large enough to avoid.

Following up on two of Corey Wall's comments: 5) I also suggest that the optimal methods for the total procedure should be more clearly laid out in the text. What do the authors suggest is the optimal sample size, leaching reagents and duration, mortar and pestle powder size, etc.

6) I had a similar though that HBr chemistry could be a useful column method that would eliminate the need for acid conversion. The procedure introduced by Amelin in 2008 has been used for U-Pb dating of trace phases (rutile, apatite). One potential issue is that I don't believe its efficacy for Sn separation has been tested.

Results: 7) This comment is not meant as a suggestion for this paper, but just a thought

for future work: I find it a compelling argument that the heterogenous Pbc composition found in C2 is likely not from inclusions of local igneous detritus, based on the ∼0.8-0.9 values of 207/206 measured in Cornubian Batholith. However, this could be tested by measuring the Pb composition of the HF after one or more cassiterite leaches.

Discussion: 8) In the beginning of the paper you describe SPG and Jian-1 as possible reference materials. I think it would be useful to come back to these samples in the conclusions, and to mention that in light of your new data these possible standards are now better characterized.

9) I think that the results of this manuscript show that both ID TIMS and LA ICP-MS have their advantages and disadvantages when it comes to cassiterite U-Pb dating. The data presented here clearly show the power of better single data point precision for resolving potentially large source of bias resulting from Pb isotope heterogeneity. However, the course sampling for ID TIMS incorporated more variable Pbc domains and hinders the ability to carefully screen domains within a single cassiterite grain. I suggest including a sentence or two in the discussion or conclusions that acknowledges this and suggests careful consideration to both the pros and cons of ID TIMS and LA ICP-MS analyses that were outlined in this paper.

Technical comments: Line 26: This sentence, while important, can be stated more clearly. Line 28: Should be more specific in regard to what "analyses" you are referring to. The x-axis in figure 7 is wrong. Line 32: While pedantic, I think you can be a little more specific than "properly characterize" to more clearly state the main conclusion. Line 68: There are two periods at the end of this sentence. Line 95: "Terra" should be spelled Tera. Line 148: "have been" is written twice. Line 242: I think it would be useful to mention here will touch more upon HF leaching below. Line 243: This sentence is unclear and should be reworded. Line 260: Consider rewording this sentence for readability Line 277: Run-on sentence. Line 287: Erroneously placed parenthesis after survey Line 415: The sentence starting here is broken. Generally, this paragraph can be reworked for readability. Line 439: The age for C1.1 here is 285.13, but in figure

Line 440: Is 0.8760 the upper intercept of the Tera-Wasserburg or the y-intercept? The same question applies to the upper intercepts reported in figure 7. Line 609: I think there should be a "to" in between "attributed variable" Line 623: I don't see a figure 9 in Moscati and Neymark 2019. Do you mean figure 5? Line 696: "cam" should be can Figure 5: On the second line you refer to Z1 and 2 instead of Z1 and Z2. On the fourth line of the caption add (Z2) after rhyolite porphyry dykes. Figure 7: Ratio on the x-axis is flipped

I hope these suggestions are helpful, Sincerely, Gavin Piccione

---

## Author Comment (AC1) · 10 May 2020

Simon Richard Tapster and Joshua William George Bright

simont@bgs.ac.uk

gchron-2019-22 Response to reviewers: "High-precision ID-TIMS Cassiterite U-Pb systematics using a low-contamination hydrothermal decomposition: implications for LA-ICP-MS and ore deposit geochronology"

We thank the reviewers for their positive comments and constructive feedback and recommendations of minor edits. We have broken reviewer comments in to some

general themes are first broken down into some more general sections/topic areas with our responses below corresponding comments. Amendments to technical comments are detailed after.

Theme 1: Tackling the "geological problem"

RC1: however, by the end I was left a little confused by the impact the results have on the overall interpretation of the geologic problem Although the cassiterite age falls in this "age window", there are too many variables associated with this age that it becomes difficult to put geologic implications or a direct process (i.e. cooling of the granitic stock and precipitation of cassiterite). We know roughly what age the cassiterite has to be, but it would be nice to directly link it to a geologic process, and unfortunately it seems impossible at this time.

Response: Although the MS is not focused on pushing geological interpretations we argue that even at the level of total uncertainty stated (including the caveats about initial disequilibrium effects) that we do answer a question about geological process in the specific example of Cligga Head presented. This question is whether the Sn ore relates to a hydrothermal system linked to magmatic activity of the host stock or the dyke ~2 Myrs later? This requires a resolution <1%. In our study the cassiterite date defines a much closer temporal association with the former thus ruling out the later as the causative generation of magmatism. We discuss how this relates to expected longevity of magmatic hydrothermal systems and geological process in section 8.2. The ability to answer higher resolution questions, such as the timing of cassiterite within a single hydrothermal system or cooling of a granite stock will be a function of: 1) the age of the cassiterite; 2) the measurement precision (a function of age and U concentrations); 3) isotopic complexities and their interpretation (Pbc concentrations, initial disequilibrium and variation, Pb loss?). We identify the key current limitations to accurately interpreting high analytical precision cassiterite data that will be applicable in future studies i.e. unconstrained disequilibrium issues. To progress beyond these levels of uncertainty (higher resolution geological questions), future studies will have to

address these issues.

RC 1: Based on this, are there any other phases you could determine the Pb isotopic composition of? What about the inclusions in the cassiterite that you dissolve during the HF leaching step?

I think there is an interesting study looking at the Pb isotopic composition in these low T hydrothermal systems

RC2: 7) This comment is not meant as a suggestion for this paper, but just a thought for future work: I find it a compelling argument that the heterogenous Pbc composition found in C2 is likely not from inclusions of local igneous detritus, based on the _0.8-0.9 values of 207/206 measured in Cornubian Batholith. However, this could be tested by measuring the Pb composition of the HF after one or more cassiterite leaches.

Response: There are numerous hydrothermal phases in the deposit such including sulfides and silicates (the mineral list is provided in section 7.1 and readers are referred to the more detailed deposit specific references for further details on mineral paragenesis). We need to ask what the real relationship between the cassiterite is to other phases beyond simply being in the same vein. As described in the discussion evidence from O isotopes indicates large variations in the hydrology (sources) of the hydrothermal system within single crystals, it could well be that minerals other than cassiterite formed within the same vein at different T and pH will show contributions from different Pb source components and in different amounts – with less "radiogenic" Pbc – and could show different Pbc isotopic compositions over timescales that are much shorter than those resolvable by radio-isotopic data.

Inclusions within cassiterite could be one way to circumvent these issues as we known these phases were present at the time of cassiterite growth. Although not included within this study, we can confirm we attempted isotopic analyses HF and Aqua Regia leachates, but failed to get any reasonable ionisation of Pb following chemical separation using the same HCl based chemistry we describe for cassiterite (we expect this

was due to matrix effects). In the future, this could be an interesting avenue to follow using different chemistry procedures to deal with issues of more complex matrices. The other way to do this would be to target mineral inclusions using a very careful approach using LA-ICP-MS or ion microprobe to gain high spatial insights into the Pb composition of inclusions.

We agree it would make an interesting study to examine how Pb isotope composition varies in space and time within the hydrothermal system, this could be linked to additional criteria such as stable O and H isotopes and fluid inclusions. This would provide a greater test of the explanatory model presented in the discussion that calls upon the high temperature greisen fluids to correspond to the cassiterite growth and the incorporation of metasediments alteration to derive radiogenic initial Pb. We think this nature of study is something for the future and beyond the scope of this MS, which focuses on U-Pb isotope systematics.

Theme 2: Defining the General Workflow

RC1: I would like to see a final figure that describes the finalized workflow the authors have determined to be the most effective way to dissolve cassiterite so that labs looking to adopt this method have something to follow. A modification of Figure 6A) would be nice to illustrate the full procedure. Do the authors think a pestle and mortar approach to cassiterite is a good option, or should individual grains or fragments of grains be analyzed? This is something that should be addressed in the conclusion

RC2: 4) It is likely that variation in initial Pb composition in a single grain of cassiterite would be unavoidable, but could you decrease the risk of sub-grain scale heterogeneity (like that in C1) by imaging (SEM or CL) the grains before ID TIMS analyses? I acknowledge that most inclusions would be much smaller than the ID TIMS sampling size, but largescale features like secondary growth zones, imbedded grains, or secondary mineral growth may be large enough to avoid.

RC2: Following up on two of Corey Wall's comments: 5) I also suggest that the optimal

methods for the total procedure should be more clearly laid out in the text. What do the authors suggest is the optimal sample size, leaching reagents and duration, mortar and pestle powder size, etc.

Response: We agree that a workflow figure would be a nice addition and would bring clarity to issues that both reviews have picked up on. As a result we have added a new schematic "work flow" figure 3 to be placed within the methods section. This figure is attached.

For wider reference this also includes potential techniques that could be employed prior to ID-TIMS U-Pb analyses to define petrogenetic characteristics of the cassiterite that may be linked to ages. It is unlikely there are mixed age domains given the nature of the material (it's effectively a hydrothermal mineral and so would be difficult to imbed grains of an older system, but overgrowths later within the same system could occur), but we agree with Reviewer 2 future work should start with CL imaged cassiterite (or other imaging techniques such as optical microscopy on fluid inclusion wafers for example). However, we feel the issues are more nuanced (sample and question specific) than Reviewer 2 suggests regarding prescription of experimental conditions. The criteria Reviewer 2 describes depend on range of issues that include: 1) what the analyst is trying to achieve (spatial resolution); 2) true age of the sample (dynamic ratios – note we ended up heavily underspiking/using too much material for SPGIV by simply using the same amount of cassiterite); 3) composition of the sample (U and Pbc concentrations and other elements that may affect decomposition time for a given grain size).

We have demonstrated that the method we present worked, and we describe what we think are the main issues to be considered when attempting this approach, but in many instances we haven't fully experimentally examined the optimal conditions, so we would prefer not to prescribe the conditions we used to future users (see responses below for further details). But we have added the parameters we used have been added to the work flow figure as a guide for people to consider.

Theme 3: Defining the leaching method

RC1: Line 243 - The authors mention an HF leaching step; however, this leaching step is not described anywhere in this preparation step. Is it the 120C overnight in 29M HF? If it is then explicitly say that and if it was in Parr vessels or on the hotplate. This is a paper outlining the proper way to digest cassiterite, be very specific about these details

Response: "On a hot plate" has been added – also see additional work flow figure and rewritten methods section with full details.

RC:1 Line 424 - Were these HF leached as well? This treatment that the various cassiterite grains received are not well documented here, clarification is needed

Response: We have added that these are HF leached. We have also modified the sub section heading 7.4.3 to reflect this.

RC2: 2) I have several questions regarding the HF leaching procedure: How long do you leach in HF (you might say this on line 242 but it's not clear that you mean that you refluxed samples in HF overnight)? How much HF do you use? Do you rinse after HF leaching? Do they lose any material after leaching? I think the paper could benefit from a clearer description of the HF leaching process.

Response: The methods section has been edited to be more explicit and notes have been added to the new work flow figure. Please see comment below for further information.

RC2: 3) You mention that the HF procedure effectively removes Pbc from inclusions that are exposed during powdering with mortar and pestle. Do you feel that this is 100% affective? Although it appears to be on C1.1, could micro-inclusions or variable levels of leaching be responsible for the differing Pbc composition in C1.1-4?

RC1: Line 436 - do the authors think this 7 ng blank for the one analysis is an inclusion?

Response: The efficacy of crushing and leaching steps in terms of comminution size,

leach time and leaching reagents will be dependent on the mineralogy of inclusion phases, their size ranges, and their spatial distribution through the cassiterite material. The type of the pestle and mortar used should also be considered as any aggregates need to be dissolved. As a result the method is not a "one size fits all" in terms of sample material and we do not want to present it as such. It is open to future users to determine what will work best for their samples. Other acids could be used if they are oxidising and will not decompose the cassiterite matrix and disturb matrix isotope systematics.

The key point we highlight based on experimental data is that HF leaching is preferable to not leaching, and we expect this to be the case regardless of sample material. We simply follow the rationale that silicates can easily be dissolved using HF but the exact and optimal conditions have not been defined, or refined, experimentally. We did do an ad hoc analysis by examining pre-leach and post-leach cassiterite on the SEM- dump mounting material on to carbon tape in partial vacuum mode, we were satisfied that we had removed aggregates from crushing in that instance. But this was not systematic analysis and relies on finding fragments of Silica at the scale of $\sim$1 um or less in amongst $\sim$20 um grains of crushed cassiterite. We are hesitant that this methodology can be used to quantify the efficacy of leaching. Stronger evidence can be taken from the low amounts of Pbc that can be achieved post-leach.

It is not possible to discriminate between the different Pbc components (inclusions, crushing aggregates) from lattice bound Pbc within the experimental setup we present. The C1.1-4 aliquots were not leached by HF, we expect the variable composition to be derived from components of blank introduced from crushing (pestle and mortar) and from inclusions, either could account for the 7 ng blank, although we think an inclusion is most likely. We think it is a simple enough extrapolation of our discussion that if inclusions are not exposed at surface, or on fractures exposed to the surface after crushing (enclaves) then these will be left within the analyte volume upon decomposition and will contribute to the Pbc. This is why considering the size and distribution of inclusions

can help a user determine their best practise. We don't think it's necessary to spell this out within the text though beyond what is present.

As stated above the methods used have been better identified within the methods and work flow figure, but we do not wish to take the hones off users to design their experiment based on their own material, based on the concepts we provide.

Theme 4: Chemistry procedures

RC1: Line 267 - Did the authors consider an HBr based chemistry for Pb? This would remove the HCl conversion step and reduce the laboratory blank slightly.

RC2: 6) I had a similar though that HBr chemistry could be a useful column method that would eliminate the need for acid conversion. The procedure introduced by Amelin in 2008 has been used for U-Pb dating of trace phases (rutile, apatite). One potential issue is that I don't believe its efficacy for Sn separation has been tested

Response: Of course there always is scope for further evaluating and optimising the different chemical separations and ionisation efficiencies! Although not explicitly stated within the MS we did also evaluate a mixed HBr-HNO3 chemistry for dealing with complex matrices, but found some component of Sn interfering with the Pb peak, although we do not know how this impacts ionisation efficiency (it could be a penalty or a benefit). With a larger volume of more complex reagents needed by the HBr-HNO3 scheme (likely a higher blank) we decided the method presented was the simplest and most effective, with no evident cost to ionisation compared to a typical zircon run.

We can't comment on the efficacy of an HBr-HCl scheme as this was not undertaken. We can say, although possibly lab specific, that we achieve over an order of magnitude cleaner 6N HCl than the UPA HBr reagent that can be commercially acquired. We typically measure <5 fg/ml Pb from HCl distillations. So the use of < 1ml of HCl for chloride conversion and resin conditioning steps is negligible, and we think is either equal to, or potentially preferable to, using 1 M HBr on the columns but skipping a step.

Whilst we completely agree lower blanks are always better, any marginal benefits to the chemistry blank that may be achieved beyond this scheme are likely to be negated by the higher initial Pbc component of cassiterite compared with material such as zircon.

Theme 5: Other method based comments

RC1 Line 255 – do you worry about the longevity of the PFA microcapsules? If there is degradation of the capsules at slightly higher temperatures do you worry about exposure to HBr over longer periods of use? These capsules can be very time consuming to clean before initial use

Response: Yes we have some concerns, but note degradation also occurs during Teflon exposure to HCl at high temperatures for extended periods of time. We thought it was responsible to note our observations to avoid users using their cleanest newest zircon capsules without understanding the potential impacts of using HBr – as the reviewer notes these are time consuming to clean. We haven't experimentally derived any data on long term usage of HBr with Teflon at these temperatures so cannot speak to the long term impact. We have suspicions that PTFE capsules would be preferable to use with HBr at high P-T conditions but have not fully demonstrated this experimentally so it's too early to say this within the MS.

RC2: Specific Comments: Methods: 1) To justify that they achieved full decomposition they use SEM and EDS scans of post HBr dissolution solids and show that there are no Sn-oxides left. Why don't they include these data? Given that full decomposition is of fundamental importance to this paper, I suggest that you include these data if not in the main text, in the supplement.

Response: It is not the case that we use this to demonstrate full decomposition of cassiterite. As described in section 4 the image of SnBr crystals (taken after the very first attempt at decomposition with HBr within the decomposition "rate" experiments) are provided as an example that the mass reduction we observed, occurs due to the chemistry we propose - HBr decomposing the cassiterite to form the Sn - Br solution

that precipitates Sn x Br y crystals upon drying (currently we are unclear on the exact formula).

Evidence of full decomposition within the later geochronology experiments was provided by optically checking the capsules under a microscope, and from extrapolating the experimental data for mass decreases over time. We can't see the additional value to including EDS spectra that simply show the images of crystals contain Sn and Br beyond what is achieved by stating this within the text. If we felt it was more pertinent and needed any other level of interpretation then we would have included the spectra.

RC1: Line 291 - What decay constant and isotopic compositions for U were used? Since this is a paper about method development and reference materials, this information is critical.

Response: References added

Theme 6: Conclusions and limitations of cassiterite

RC1: Line 590 - a poor way to end a discussion session, try rewording this so it does not have such a negative tone to it. Maybe delete this sentence and end with the previous sentence.

Response: We disagree with reviewer in this instance. This study is about defining the current limitations of cassiterite U-Pb geochronology, as much as it is about generating new possibilities with this method. This is why the title refers to "high precision U-Pb systematics" not high precision ages or dating. Understanding the components of uncertainty, beyond those simply derived from isotopic measurements and data reduction outputs is fundamental to how we interpret geological processes based on our isotopic dates. This sentence is critical to any future research that presents high-precision ID-TIMS cassiterite dates, and indeed the limits of interpretations for LA-ICPMS high-n data sets. We don't think it should be removed.

RC1: Line 606 - this is a very troublesome sentence, how do the authors suggest

combating this variable Pbc mixing or is it a problem that can be solved? If the latter is the case, then it severely hinders the interpretation of cassiterite U-Pb ages. .

Response: Yes, it limits the interpretation. Identification of this limitation to the precision of date interpretation is in many ways the point of this MS, whilst a hindrance to the precision of cassiterite dates, we are promoting accuracy (at the appropriate level of precision).

The isotopic systematics of sample material limit how far microbeam data sets can be interpreted due to their single data point uncertainty. Accuracy of date interpretations are not down to mass spectrometry or ablation issues alone. We are highlighting that by extending the interpretation of data sets beyond what is laid out in this MS through use of high-n analyses the interpreted dates may be inaccurate at their stated level of precision even if, hypothetically speaking, analytical protocols were perfect. The same reasoning can be applied to ID-TIMS data sets, just at a different resolution. We present a strategy of how to over-come this issue with ID-TIMS sampling by generating an "internal isochron" by analyses of the same parental batch of crushed material. If there is not any detectable variation between the evident Pbc composition between data from different parental fragments, there is no reason why data from spatially different parts couldn't be combined within the date interpretation, but we are likely to introduce dispersion that limits the accuracy of dates where potentially different Pbc ICs are combine and where high precision measurements start to resolve that, as is evident in our results.

We would suggest that for microbeam techniques one potential solution/strategy, to at least reduce the impact of variable Pbc IC, would be to utilise spot patterns that follow growth zones (evident in CL) rather than traverse them. The caveat is that this would reduce the range of U-Pb elemental variation and therefore limit the capacity to constrain regression lines through data, and ultimately reduce the precision of a lower intercept date, although this would be traded off with accuracy. It will all depend on how large the variation in Pbc IC is and over what spatial resolution it varies.

RC1: Line 614 - Again, any suggestions on how to correct for this?

Response:Sampling strategies for ID-TIMS are outlined depending on technique used but this is a limiting uncertainty on the date interpretation.

RC1: The conclusions section needs to be reworked. The authors did a great job characterizing the issues with U-Pb cassiterite geochronology and make great strides forward in this field, but the last paragraph and concluding sentences of paragraphs leaves me curious of what the actual outcome of this manuscript is. The last paragraph in the conclusions should be removed entirely and replaced. The end of the abstract should be revisited by the authors to aid in writing a satisfactory conclusions section

Response: We note that it is not the scientific merit of the conclusions that the reviewer rejects. Cassiterite is being increasingly used to date ore deposits by LA, and now we have a tool for high precision measurements. It is imperative that we recognise where the limitations come in interpreting "ages" for geological events from our isotopic dates so we can derive accurate geological models at the various temporal resolutions different methodologies permit.

The conclusion we present is - that by using the high precision isotopic measurements we have defined limitations to microbeam data interpretation. Therefore limitations on the accuracy of interpreted dates beyond a certain level of single data point precision is a valid one, and we would argue the most important conclusion of the MS for the geological interpretations based on Cassiterite U-Pb geochronology. Whilst this may not be the conclusion everybody wants to hear, and I agree it would be preferable to geochronology if this wasn't present, we should not try to bury this issue.

RC2: Discussion: 8) In the beginning of the paper you describe SPG and Jian-1 as possible reference materials. I think it would be useful to come back to these samples in the conclusions, and to mention that in light of your new data these possible standards are now better characterized.

Response: The conclusions state "Comparison of ID-TIMS and previously reported LA-ICP-MS U-Pb datasets for the three samples indicate that the latter are accurate at the ∼1 % (2σ) level, especially when taking into account the ID-TIMS date that was ∼0.4 % younger than the inferred age for the SPG-IV cassiterite used to normalise matrix effects in the Neymark et al. (2018) study. This direct U/Pb determination can be used to refine LA-ICP-MS data that are normalised to this ad hoc reference material (note we only use this is the sense that it has been used before rather than defining it as a reference material ourselves), further improving the underpinning of cassiterite U-Pb via LA-ICP-MS.

We deliberately avoided stating "reference values" within the conclusions or elsewhere, because of variable amounts of initial Pbc the reference value will be generated from a reference line (regression) and any dispersion on this line. Despite the data we present offering a marked improvement we would be extremely hesitant to recommend values that appear to be fully representative on the basis of 3-4 data points. We see the data presented much more as an interim step in achieving a well characterised RM for cassiterite.

Response to SC1 (Norman): No revision needs to be made in this instance as this method was first presented at GS 2019 from which this publication builds on, 'Tapster S & Bright J, Goldschmidt Abstracts, 2019, 3324', so we would cite this instead. Additionally the DOI of our Gchron discussion article does not follow the timeline inferred in the comment.

RC1: Technical comments

Line 12 - Change "the mineral" to cassiterite, add comma after "common Pb" / Changed Line 15 - delete "to date," / deleted Line 17 - delete "," after reference / deleted Line 26 - should be "demonstrates" / Changed Line 32 - delete "However," / Changed Line 39 - add some references at the end of the first paragraph / Added Line 46 - delete "Yet" / deleted Line 77 - delete "However, it" and replace with "This" / We disagree,

the "however" contrasts the topic of U-Pb fractionation from the previous sentence that discusses preservation of the Pb signature and rad-common mixing, these are two issues with two very different effects Line 90 - add "in-situ" between resulting and ages / Added Line 91 - This is a run on sentence, consider breaking these thoughts into two sentences / Changed Line 104 - Awkward sentence, needs reworking / Changed Line 107 - Again, a run on sentence, consider breaking these thoughts into two sentences / Changed Line 114 - delete "that" after 2) / Changed Line 115 - delete "That" after 3) / Changed Line 137- delete "and at the time of writing was" and replace with is / Changed Line 138 - comma after "Pb",delete "was" and replace "Likewise" with Similarly / Changed Line 141 - Delete "Furthermore" / Changed Line 150 - Give range for what is typically achieved for modern CA-ID-TIMS methods / Changed Line 152 - changed "indicate" to indicated / Changed Line 158 - delete "the study of" / Changed Line 159 – add comma after (1992) / Changed Line 164 - add reference Scoates and Wall, 2015. Geochronology of Layered Intrusions / Reference Added Line 167 - Delete "yet" before "been" / Changed Line 183 - delete comma after Br2 / Changed Line 221 - delete "before the HBr is lost and becomes ineffective" / Not deleted but further clarification added Line 439 - delete "Together they yield" and replace with "and yielded" / Changed Line 655 - delete "As" / Changed Line 696 - replace "cam" with "can" / Changed

RC2: Technical comments:

Line 26: This sentence, while important, can be stated more clearly. / No change Line 28: Should be more specific in regard to what "analyses" you are referring to. / Changed to "data" Line 32: While pedantic, I think you can be a little more specific than "properly characterize" to more clearly state the main conclusion. / In its current form we think this accurately describes the stage at which our experiment has taken us to. As described above, we cannot predict the results or review every permutation of all cassiterite materials that could be analysed. We would like this to stimulate better characterisation in the future. Line 68: There are two periods at the end of this

sentence. / Changed Line 95: "Terra" should be spelled Tera. / Changed Line 148: "have been" is written twice. / Changed Line 242: I think it would be useful to mention here will touch more upon HF leaching below. / See previous comments on HF leaching Line 243: This sentence is unclear and should be reworded. / This section has been reworked with the addition of further details Line 260: Consider rewording this sentence for readability / Changed Line 277: Run-on sentence. / Changed Line 287: Erroneously placed parenthesis after survey / Changed Line 415: The sentence starting here is broken. Generally, this paragraph can be reworked for readability./ Changed Line 439: The age for C1.1 here is 285.13, but in figure - / This comment seemed incomplete. We assume the reviewer identified a discrepancy in the date between the results and the figures, this was down to a mistyping within the text and has been edited to 285.14 as per figures (and data reduction outputs) Line 440: Is 0.8760 the upper intercept of the Tera-Wasserburg or the y-intercept? The same question applies to the upper intercepts reported in figure 7. / Thanks for pointing this out, we are referring to the y-intercept and for clarity we have edited this accordingly and elsewhere when this issue occurs to either "y-intercept" or "207Pb/206Pb intercept" Line 609: I think there should be a "to" in between "attributed variable" / Changed Line 623: I don't see a figure 9 in Moscati and Neymark 2019. Do you mean figure 5? / This referred to our Figure 9, and the sentence has been modified for clarity Line 696: "cam" should be can / Changed Figure 5: On the second line you refer to Z1 and 2 instead of Z1 and Z2. On the fourth line of the caption add (Z2) after rhyolite porphyry dykes. / Changed Figure 7: Ratio on the x-axis is flipped/ The x-axis in figure 7 is wrong. / Corrected

[Figure]

**Sampling for cassiterite high-precision ID-TIMS date "internal isochron" from single parent fragment**

**Sampling to test spatial variation of U-Pb isotope systematics by ID-TIMS e.g. Reference Material characterisation**

**Potential pre-ID-TIMS U-Pb petrogenetic characterisation**

Vein petrography

BSE + CL imaging

LA-ICPMS Trace elements and mapping

Oxygen isotopes

Fluid Inclusions (wafer sample)

Mineral inclusion analyses

Microbeam U-Pb Screening/mapping

Crush + homogenise cassiterite (<20-50 μm) using agate pestle and mortar under acetone expose inclusions

*Transfer and Rinse ($H_2O$)*

Leach Stage 1
e.g. 120°C hotplate overnight Aqua Regia Targeting removal of sulfides, phosphates, oxides

*Rinsing ($H_2O$ or 4M $HNO_3$)*

Leach Stage 2
e.g. 120°C hotplate overnight 29M HF +trace $HNO_3$ Targeting removal of silicates and other residues

*Rinsing ($H_2O$ or 4M $HNO_3$)*

Sub-sample amount required per aliquot/analysis depending on age, blank, U conc, decomposition rate e.g. 0.1 mg to 1 mg

Transfer to capsule and rinse(4M $HNO_3$ + 1M HBr) Isotopic tracer addition to microcapsule

Decomposition in >40x excess 9M HBr 200-220°C HP vessel (time is mass, sample and grain size dependent)

Chloride conversion

Column Chemistry (HCl based AG1x8)

*Pb Elution*          *U Elution*

Column Chemistry (HCl-$HNO_3$ AG1x8) U-clean up Fe removal

TIMS Analyses

**Fig. 1.** New workflow figure (new Figure 3)

---

## Referee Report (RR1)

In their revisions, Tapster and Bright have added clarifying information to their methods sections, along with changes to the text to improve the flow of the manuscript. Section 5 has been significantly improved, which now lays out the methodological workflow and leaching technique in a clearer way. Figure 3 is a really nice addition that I think will be very useful for researchers looking to utilize methods described in this paper. Authors have done a nice job of making technical improvements to the text.

In the interactive discussion for this manuscript there was some discussion about the geologic implications of the U-Pb ages for the Cligga Head greisen deposit. It is my opinion that the authors demonstrate a careful analysis of the implications of these data without over interpreting them. Specifically, their discussion in the paragraph beginning on line 606 does a nice job presenting the geologic conclusions, as well as the uncertainties preventing further conclusions.

In their response to reviewers, the authors also justify the contents of their conclusions section. I feel that the conclusions section effectively presents the findings of the paper.

It is my opinion that the revised iteration of the manuscript is suitable for final publication in Geochronology, following the few technical changes I suggest below.

Technical Comments:
Line 32: The last sentence in the abstract seems to transition abruptly from the previous sentence describing the inaccuracy of microbeam methods. I think it could help to state specifically, "the ID-TIMS methods described in this manuscript", rather than "This method".

Line 52: No need to start a new paragraph here

Line 121: fulfill spelled incorrectly

Line 273: "placing" should be "placed"